# SCALABLE NEURAL NETWORK KERNELS

**Arijit Sehanobish**[*]
Independent Researcher

**Krzysztof Choromanski**[*]
Google DeepMind &
Columbia University

**Yunfan Zhao**[*]
Harvard University

**Avinava Dubey**[*]
Google Research

**Valerii Likhosherstov**[*]
Waymo

## ABSTRACT

We introduce the concept of *scalable neural network kernels* (SNNKs), the replacements of regular *feedforward layers* (FFLs), capable of approximating the latter, but with favorable computational properties. SNNKs effectively disentangle the inputs from the parameters of the neural network in the FFL, only to connect them in the final computation via the dot-product kernel. They are also strictly more expressive, as allowing to model complicated relationships beyond the functions of the dot-products of parameter-input vectors. We also introduce the *neural network bundling process* that applies SNNKs to compactify deep neural network architectures, resulting in additional compression gains. In its extreme version, it leads to the fully bundled network whose optimal parameters can be expressed via explicit formulae for several loss functions (e.g. mean squared error), opening a possibility to bypass backpropagation. As a by-product of our analysis, we introduce the mechanism of the *universal random features* (or URFs), applied to instantiate several SNNK variants, and interesting on its own in the context of scalable kernel methods. We provide rigorous theoretical analysis of all these concepts as well as an extensive empirical evaluation, ranging from point-wise kernel estimation to Transformers' fine-tuning with novel adapter layers inspired by SNNKs. Our mechanism provides up to 5x reduction in the number of trainable parameters, while maintaining competitive accuracy.

## 1 INTRODUCTION

Consider a kernel function: $\mathrm{K} : \mathbb{R}^d \times \mathbb{R}^d \to \mathbb{R}$, taking as input two feature vectors encoding latent embeddings of their corresponding objects and returning their similarity. Kernel methods are among the most theoretically principled approaches to statistical machine learning (ML) and have proven effective in numerous real-world problems (Schölkopf & Smola, 2002; Kontorovich et al., 2008). Despite their theoretical guarantees and applicability in a rich spectrum of ML settings, the main drawback of these techniques is a high computational complexity, at least quadratic in the size $N$ of the training dataset. For example, the kernel regression has time complexity $\mathcal{O}(N^3)$.

To address this issue, Rahimi & Recht (2007) proposed to construct a random feature (RF) map $\Phi : \mathbb{R}^d \to \mathbb{R}^m$ that transforms an input point $\mathbf{z}$ to a finite-dimensional feature vector $\Phi(\mathbf{z}) \in \mathbb{R}^m$ such that: $\mathrm{K}(\mathbf{x}, \mathbf{y}) = \mathbb{E}[\Phi(\mathbf{x})^\top \Phi(\mathbf{y})]$ (effectively approximately linearizing kernel function). Approximating general kernels $\mathrm{K}(\mathbf{x}, \mathbf{y})$ via linear (dot-product) kernels $\mathrm{K}(\mathbf{x}, \mathbf{y}) \approx \widehat{\mathbf{x}}^\top \widehat{\mathbf{y}}$ for $\widehat{\mathbf{z}} = \Phi(\mathbf{z})$ drastically changes computational complexity landscape, which is now dominated by the number $m$ of random features, thus providing computational gains if $m \ll N$. Since their seminal work, there has been a variety of works proposing random features for a broad range of kernels like Gaussian, Matern (Choromanski et al., 2018) and polynomial (Kar & Karnick, 2012; Wacker et al., 2022a;b).

In the meantime, with the advances in: the optimization algorithms for deep ML architectures and the accelerators' hardware, neural network (NN) models (Goodfellow et al., 2016; Schmidhuber, 2014; LeCun et al., 2015) have become predominant in machine learning.

---

[*]Equal Contribution

The *feedforward layer* (FFL) is the core computational module of NNs and is of the following form:

$$\mathbf{x} \to f(\mathbf{W}\mathbf{x} + \mathbf{b}) \qquad (1)$$

for $\mathbf{x} \in \mathbb{R}^d, \mathbf{W} \in \mathbb{R}^{l \times d}, \mathbf{b} \in \mathbb{R}^l$ (*bias*) and an *activation function* $f : \mathbb{R} \to \mathbb{R}$ (applied element-wise). The expressiveness of deep NNs, far surpassing standard kernel methods, comes from stacking together several FFLs, each encoding **non-linear** mapping with **learnable** $\mathbf{W}, \mathbf{b}$.

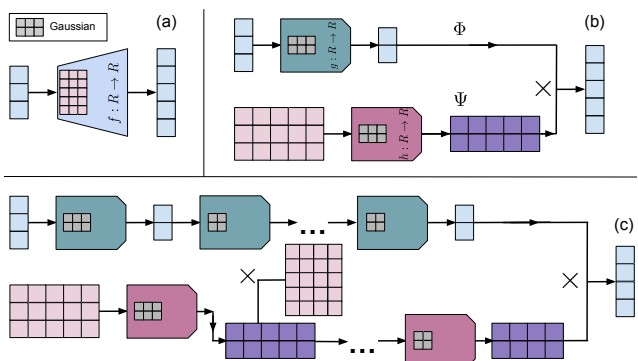

Figure 1: Pictorial representation of different NN layers discussed in the paper. Pink arrays represent NN weight matrices and grey ones, Gaussian projections matrices applied in SNNKs. Nonlinear transformations applied in mappings $\Phi$ and $\Psi$ are symbolically represented as functions $g$ and $h$ respectively. **Upper left:** Regular FFL with activation $f$. **Upper right:** SNNK applied to a single FFL. **Bottom:** Bundling process using SNNKs and applied to a deep neural network.

In this work, we draw a deep connection between scalable kernel methods and neural networks. We reinterpret the FFL as outputting the expected vector of dot-products of: **(1)** the latent embeddings of the input $\mathbf{x}$ and **(2)** the parameters: $\mathbf{W}, \mathbf{b}$ of the FFL, effectively disentangling input from model's parameters in the computational graph, only to connect them in the final computation via the dot-product kernel. To be more specific, we think about the FFL as the following transformation:

$$\begin{cases} \overline{\mathrm{K}}_f(\mathbf{x}, (\mathbf{W}, \mathbf{b})) \stackrel{\text{def}}{=} \left( \mathrm{K}_f(\mathbf{x}, (\mathbf{w}^0, b_0)), ..., \mathrm{K}_f(\mathbf{x}, (\mathbf{w}^{l-1}, b_{l-1})) \right)^\top, \\ \mathrm{K}_f(\mathbf{x}, (\mathbf{w}, b)) \stackrel{\text{def}}{=} \mathbb{E}[\Phi_f(\mathbf{x})^\top \Psi_f(\mathbf{w}, b)], \end{cases} \qquad (2)$$

where mappings: $\Phi_f : \mathbb{R}^d \to \mathbb{R}^m, \Psi_f : \mathbb{R}^d \times \mathbb{R} \to \mathbb{R}^m$ satisfy: $f(\mathbf{w}^\top \mathbf{x} + b) = \mathbb{E}[\Phi_f(\mathbf{x})^\top \Psi_f(\mathbf{w}, b)]$ and $\mathbf{w}^0, ...\mathbf{w}^{l-1}$ are the transposed rows of $\mathbf{W}$. Then, in the instantiation of the layer the expectations are dropped out. Rewriting an FFL in terms of two towers: one corresponding to the input and one to its learnable parameters has several advantages:

1. **network compression:** in the above formulation, instead of transforming layer parameters with $\Psi_f$, one can directly learn vectors $\Psi_f(\mathbf{w}^i, b_i)$ for $i = 0, ..., l-1$. Then the number of trainable parameters becomes $O(lm)$ rather than $O(ld)$ and for $m \ll d$ the layer effectively has a reduced number of parameters.

2. **computational savings:** if RFs can be constructed in time $o(dl)$ per point and $m \ll d$, the overall time complexity $o(dl)$ of the FFL (given pre-computed embeddings $\Psi_f(\mathbf{w}^i, b_i)$) is **sub-quadratic** in layers' dimensionalities,

3. **deep NN bundling process:** a two-tower representation can be used iteratively to compactify multiple FFLs of NNs, the process we refer to as *neural network bundling* (Sec. 3.3); this also leads to the computational gains.

4. **deep NNs as scalable kernels:** the extreme version of the bundling procedure, involving all the layers, provides a two-tower factorization of the entire deep NN with several potential practical and theoretical implications (Sec. 3.3). In particular, it leads to an explicit formula for the optimal parameters of the fully-bundled network under several loss objectives (e.g. mean squared loss), opening a possibility to bypass backpropagation.

In order to find mappings: $\Phi_f, \Psi_f$ from Eq. 2, we develop a new bounded random feature map mechanism, called *universal random features* (or URFs) that leads to the unbiased estimation of $f(\mathbf{w}^\top \mathbf{x} + b)$ as long as $f$ has a well-defined Fourier Transform (FT), either in the classical Riemannian or distributional sense. To derive URFs, we combine Fourier analysis techniques with recent methods for softmax-kernel estimation from Likhosherstov et al. (2022).

**Note:** We do not put any additional assumptions regarding $f$, in particular $f$ is not required to be differentiable. Furthermore, function $\mathrm{K}_f$ **does not need** to be positive semi-definite. This is critical for applications in neural networks, where the activation function $f$ usually does not correspond to a positive semi-definite kernel.

To summarize, our main contributions in this paper are as follows:

- We introduce the *scalable neural network kernel* module (SNNK) as a replacement of a traditional FFL (Sec. 3), providing the disentanglement of the network's input and its parameter-set before final dot-product computation, as given in Eq. 2 (see also: Fig. 1).
- We accompany SNNKs with our universal random features mechanism (URFs) to efficiently: (1) construct mappings $\Phi_f$ and $\Psi_f$ from Eq. 2 and consequently: (2) implement SNNKs (Sec. 3.1). We provide explicit formulae for URFs for trigonometric maps. Those produce SNNK-based replacements of the SIREN networks from Sitzmann et al. (2020).
- We propose new NN-layers corresponding to the specific SNNK instantiation, called ReLU-SNNK (Sec. 3.2), that we found particularly effective in downstream applications (see: Sec. 4.3.2). We show that they are related to the class of the *arc-cosine kernels* (Cho & Saul, 2011). We also demonstrate using them that SNNKs are **strictly more expressive** than regular FFLs, as allowing to compute the functions of the inputs and parameters that cannot be defined as point-wise transformed vectors of their dot-products.
- We introduce the neural network compactification process, that we refer to as *neural network bundling*, leveraging SNNKs (see: Sec. 3.3 and Fig. 1).
- We provide an exhaustive empirical evaluation of SNNKs, from point-wise kernel estimation to the adapter-based Transformers' fine-tuning, providing about 5x reduction of the number of trainable parameters (Sec. 4).

Commonly used methods for compressing neural networks are pruning (Liang et al., 2021), distillation (Gou et al., 2021) and quantization (Gholami et al., 2021). Compressing neural networks via SNNKs is novel and completely orthogonal to these methods and can be combined with such.

## 2 RELATED WORK

The literature on random features is vast, yet most of the works focus on approximating positive definite kernels. The results on dimensionality reduction and the so-called *Johnson-Lindenstrauss Transform* (or JLT) (Dasgupta & Gupta, 2003; Dasgupta et al., 2010; Ailon & Liberty, 2013) for the dot-product kernel marked the birth of the subject as as an archetype mechanism that Rahimi & Recht (2007) extended from linear to non-linear shift-invariant kernels. A substantial effort was made to further improve the accuracy of RF-methods by entangling projections used to construct RFs (Choromanski et al., 2017; Yu et al., 2016; Choromanski et al., 2018; Rowland et al., 2018).

For certain classes of functions $f$, RF-mechanisms leading to the linearization of $\mathrm{K}_f$ have been already developed. In addition to the rich recent literature on the approximation techniques for the softmax-kernel $\mathrm{K}_{\exp}(\mathbf{x}, \mathbf{y}) = \exp(\mathbf{x}^\top \mathbf{y})$ (Likhosherstov et al., 2022; 2023; Choromanski et al., 2021), algorithms for analytic $f$ with positive coefficients of their Taylor series expansion were given (Kar & Karnick, 2012). Other RF-methods assume that kernel inputs are taken from the unit-sphere (Scetbon & Harchaoui, 2021; Han et al., 2022). Both assumptions are unrealistic for the neural network applications as far as inputs $\mathbf{x}$ are concerned (interestingly, the latter one would be however more justifiable for the parameter-tower as long as bounded-norm weight matrices are considered, e.g. *orthogonal neural networks* (Helfrich et al., 2018). We would like to emphasize that our two-tower mechanism, effectively leading to the linearization of the FFLs from Eq. 2, can in principle work with various RF-algorithms, and not only our proposed URFs.

The kernels applied in connection to neural networks have been widely studied (Bengio & Lecun, 2007). Such kernels are generally constructed using dot-products of outputs of the shallow neural networks with various non-linearities like ReLU (Cho & Saul, 2009; Bresler & Nagaraj, 2020) and tanh (Williams, 1996) or the gradients of the network like the NTK kernel (Jacot et al., 2020). Most of the work in linearizing NNs via kernels have been done in the case of a 2-layer network where

$$J(\mathbf{x}; \boldsymbol{\theta}) = \sum_{i=1}^{N} a_i f(\mathbf{x}^\top \mathbf{w}^i), \ \boldsymbol{\theta} = (a_1, ..., a_N; \mathbf{w}^1, ..., \mathbf{w}^N) \in \mathbb{R}^{N(d+1)} \tag{3}$$

It is assumed that $\mathbf{w}^i$ and $f$ (non-linear activation) are fixed and scalars and $a_i$ are trainable. Under various assumptions, one can write a compact linearized form of this neural network (Cho & Saul, 2009; 2011; Ghorbani et al., 2020). Moreover, in the above setting, $J(\mathbf{x}; \boldsymbol{\theta})$ corresponds to the first-order Taylor expansion of $J$ with respect to the top-layer weights $a_i$ which was first explored by (Neal, 1996). Even though our setting is fundamentally different, as our goal is to linearize single layers to disentangle the weights and the inputs, we build on the above intuition to create our SNNK-layers (see also: discussion in Appendix C). Note that NTK-based analysis, as leveraging Taylor-based linearization of the NN, is valid only for the mature stage of training/finetuning when

weights do not change much and thus such a linearization is accurate (Malladi et al., 2022). SNNKs do not rely on this assumption. Furthermore, SNNKs can be used also in the context of non-positive definite (Ong et al., 2004) and asymmetric (He et al., 2023) kernels since mappings $\Phi$ and $\Psi$ in principle are different (on expectation they can produce both symmetric and asymmetric functions).

Arc-cosine kernels were studied in the context of deep NNs before (Cho & Saul, 2009). However, in (Cho & Saul, 2009), the weights are still entangled with the FFL-input, as the initial latent representations of the inputs (for random parameters) are interpreted as RFs for the arc-cosine kernel.

## 3 SCALABLE NEURAL NETWORK KERNELS (SNNKs)

The scalable neural network kernel (SNNK) computational module is defined as follows:

$$\begin{cases} \overline{\mathrm{SNNK}}_f(\mathbf{x}, (\mathbf{W}, \mathbf{b})) \stackrel{\mathrm{def}}{=} \left(\mathrm{SNNK}_f(\mathbf{x}, (\mathbf{w}^0, b_0)), ..., \mathrm{SNNK}_f(\mathbf{x}, (\mathbf{w}^{l-1}, b_{l-1}))\right)^\top, \\ \mathrm{SNNK}_f(\mathbf{x}, (\mathbf{w}, b)) \stackrel{\mathrm{def}}{=} \Phi_f(\mathbf{x})^\top \Psi_f(\mathbf{w}, b), \end{cases} \tag{4}$$

for, $\mathbf{x} \in \mathbb{R}^d$, some mappings: $\Phi_f : \mathbb{R}^d \to \mathbb{R}^m$, $\Psi_f : \mathbb{R}^d \times \mathbb{R} \to \mathbb{R}^m$ and transposed rows of $\mathbf{W} \in \mathbb{R}^{l \times d}$: $\mathbf{w}^0, ... \mathbf{w}^{l-1}$. As we show in Sec. 3.1, functions $\Phi_f, \Psi_f$ can be constructed in such a way that the SSNK module approximates a particular FFL, i.e.: $\overline{\mathrm{SNNK}}_f(\mathbf{x}, (\mathbf{W}, \mathbf{b})) \approx f(\mathbf{W}\mathbf{x} + \mathbf{b})$, but mechanisms that do not imitate known FFLs are also of interest (see: Sec. 3.2).

**Time complexity:** If we denote by $t_m(d)$ time complexity for constructing an embedding: $\Phi_f(\mathbf{x})$, then time complexity for constructing $\overline{\mathrm{SNNK}}_f(\mathbf{x}, (\mathbf{W}, \mathbf{b}))$ (given the pre-computed $\Psi_f(\mathbf{w}^i, b_i)$ for $i = 0, ..., l-1$) is: $T_{m,l}(d) = ml + t_m(d)$. In Sec. 3.1 we show an algorithms for constructing URFs in time $t_m(d) = O(md)$ and thus computational gains are provided as compared to the regular FFL (with time complexity $O(ld)$) as long as $m \ll \min(l, d)$.

**FFL compression:** As already mentioned in Sec. 1, the key observation is that in the setting, where the layer is learned (and thus $\mathbf{w}^0, ..., \mathbf{w}^{l-1}$ are learnable), mapping $\Psi_f$ **does not even need to be applied**, since vectors $\boldsymbol{\omega}^j \stackrel{\mathrm{def}}{=} \Psi_f(\mathbf{w}^j, b)$ for $j = 0, ..., l-1$ can be interpreted as unstructured learnable vectors. Thus the number of trainable parameters of the SNNK layer is $O(ml)$, instead of $O(dl)$ and consequently, the FFL is effectively compressed if $m \ll d$.

### 3.1 UNIVERSAL RANDOM FEATURES (URFs)

In this section, we show how to construct embeddings $\Phi_f(\mathbf{x})$ and $\Psi_f(\mathbf{w}, b)$ (additional intuition is provided in Sec. A). We denote by $\mathrm{FT}_f$ the *Fourier Transform* of $f$, where $i \in \mathbb{C}$ satisfies: $i^2 = -1$:

$$\mathrm{FT}_f(\xi) = \int_{\mathbb{R}} f(z) \exp(-2\pi i \xi z) dz \tag{5}$$

If the integral does not exist in the classical Riemannian sense, we use its distributional interpretation. We rewrite $\mathrm{FT}_f$ as: $\mathrm{FT}_f = \mathrm{FT}_f^{\mathrm{re},+} - \mathrm{FT}_f^{\mathrm{re},-} + i\mathrm{FT}_f^{\mathrm{im},+} - i\mathrm{FT}_f^{\mathrm{im},-}$, where: $\mathrm{FT}_f^{\mathrm{re},+}, \mathrm{FT}_f^{\mathrm{re},-}, \mathrm{FT}_f^{\mathrm{im},+}, \mathrm{FT}_f^{\mathrm{im},-} : \mathbb{R} \to \mathbb{R}_{\geq 0}$. Without loss of generality, we will assume that all four functions are not identically zero.

Let us denote by $\overline{\mathcal{P}}_0, \overline{\mathcal{P}}_1, \overline{\mathcal{P}}_2, \overline{\mathcal{P}}_3$ some probabilistic distribution on $\mathbb{R}$ (e.g. Gaussian) and by $\overline{p}_0, \overline{p}_1, \overline{p}_2, \overline{p}_3 : \mathbb{R} \to \mathbb{R}_{\geq 0}$ their corresponding density functions. Furthermore, denote by $\mathcal{P}_0, \mathcal{P}_1, \mathcal{P}_2, \mathcal{P}_3$ probabilistic distributions of densities: $p_0, p_1, p_2, p_3 : \mathbb{R} \to \mathbb{R}_{\geq 0}$ proportional to: $\mathrm{FT}_f^{\mathrm{re},+}, \mathrm{FT}_f^{\mathrm{re},-}, \mathrm{FT}_f^{\mathrm{im},+}, \mathrm{FT}_f^{\mathrm{im},-}$ respectively. We can then write:

$$f(z) = \int_{\mathbb{R}} \mathrm{FT}_f(\xi) \exp(2\pi i \xi z) d\xi = \sum_{j=0}^{3} c_j \int_{\mathbb{R}} \frac{p_j(\xi)}{\overline{p}_j(\xi)} \exp(2\pi i \xi z) \overline{p}_j(\xi) d\xi = \sum_{j=0}^{3} c_j \mathbb{E}_{\xi \sim \overline{\mathcal{P}}_j} \left[ \frac{p_j(\xi)}{\overline{p}_j(\xi)} \exp(2\pi i \xi z) \right], \tag{6}$$

where: $c_0 = \int_{\mathbb{R}} \mathrm{FT}_f^{\mathrm{re},+}(\tau) d\tau, c_1 = -\int_{\mathbb{R}} \mathrm{FT}_f^{\mathrm{re},-}(\tau) d\tau, c_2 = i \int_{\mathbb{R}} \mathrm{FT}_f^{\mathrm{im},+}(\tau) d\tau$, and furthermore $c_3 = -i \int_{\mathbb{R}} \mathrm{FT}_f^{\mathrm{im},-}(\tau) d\tau$. For $\mathbf{x}, \mathbf{w} \in \mathbb{R}^d, b \in \mathbb{R}$, let us denote:

$$\widehat{f}_j(\mathbf{x}, \mathbf{w}, b) = c_j \mathbb{E}_{\xi \sim \overline{\mathcal{P}}_j} \left[ \frac{p_j(\xi)}{\overline{p}_j(\xi)} \exp\left(2\pi i \xi (\mathbf{x}^\top \mathbf{w} + b)\right) \right] = c_j \mathbb{E}_{\xi \sim \overline{\mathcal{P}}_j} [S_j(\xi, b) \exp(\widehat{\mathbf{x}}^\top(\xi) \widehat{\mathbf{w}}(\xi))] \tag{7}$$

for $S_j(\xi, b) = \frac{p_j(\xi)}{\overline{p}_j(\xi)} \exp(2\pi i \xi b)$, $\widehat{\mathbf{x}}(\xi) = \rho(\xi)\mathbf{x}$, $\widehat{\mathbf{w}}(\xi) = \eta(\xi)\mathbf{w}$, where $\rho(\xi), \eta(\xi) \in \mathbb{C}$ satisfy: $\rho(\xi)\eta(\xi) = 2\pi i \xi$. Inside the expectation in Eq. 7, we recognize the softmax-kernel value

$K_{exp}(\widehat{\mathbf{x}}(\xi), \widehat{\mathbf{w}}(\xi)) = \exp(\widehat{\mathbf{x}}^{\top}(\xi)\widehat{\mathbf{w}}(\xi))$. We thus disentangle $\widehat{\mathbf{x}}(\xi)$ from $\widehat{\mathbf{w}}$ there, by applying softmax-kernel linearization mechanism from Likhosherstov et al. (2022): $\exp(\widehat{\mathbf{x}}^{\top}(\xi)\widehat{\mathbf{w}}(\xi)) = \mathbb{E}_{\mathbf{g}\sim\mathcal{N}(0,\mathbf{I}_d)}[\Lambda_{\mathbf{g}}(\widehat{\mathbf{x}})\Lambda_{\mathbf{g}}(\widehat{\mathbf{w}})]$, where $\Lambda_{\mathbf{g}}: \mathbb{R}^d \to \mathbb{R}$ is defined as follows for $A \leq 0$:

$$\Lambda_{\mathbf{g}}(\mathbf{z}) = (1 - 4A)^{\frac{d}{4}} \exp(A\|\mathbf{g}\|_2^2 + \sqrt{1-4A}\mathbf{g}^{\top}\mathbf{z} - \frac{\|\mathbf{z}\|_2^2}{2}) \tag{8}$$

Thus $\widehat{f_j}(\mathbf{x}, \mathbf{w}, b) = \mathbb{E}_{(\xi,\mathbf{g})\sim\overline{\mathcal{P}}_j\otimes\mathcal{N}(0,\mathbf{I}_d)}[\Gamma^1_{\mathbf{g},\xi}(\mathbf{x})\Gamma^2_{\mathbf{g},\xi}(\mathbf{w}, b)]$ for $\Gamma^1_{\mathbf{g},\xi}(\mathbf{x}), \Gamma^2_{\mathbf{g},\xi}(\mathbf{w}, b)$ given as:

$$\Gamma^1_{\mathbf{g},\xi}(\mathbf{x}) = \Lambda_{\mathbf{g}}(\rho(\xi)\mathbf{x}), \; \Gamma^2_{\mathbf{g},\xi}(\mathbf{w}, b) = c_j S_j(\xi, b)\Lambda_{\mathbf{g}}(\eta(\xi)\mathbf{w}) \tag{9}$$

That observation directly leads to the RF mechanism for the estimation of $\widehat{f_j}(\mathbf{x}, \mathbf{w}, b)$. We can rewrite: $\widehat{f_j}(\mathbf{x}, \mathbf{w}, b) = \mathbb{E}[\Phi^j(\mathbf{x})^{\top}\Psi^j(\mathbf{w}, b)]$ for $(\xi_1, \mathbf{g}_1), ..., (\xi_m, \mathbf{g}_m) \sim \overline{\mathcal{P}}_j \otimes \mathcal{N}(0, \mathbf{I}_d)$ and:

$$\Phi^j(\mathbf{x}) = \frac{1}{\sqrt{m}}(\Gamma^1_{\mathbf{g}_1,\xi_1}(\mathbf{x}), ..., \Gamma^1_{\mathbf{g}_m,\xi_m}(\mathbf{x}))^{\top}, \quad \Psi^j(\mathbf{w}, b) = \frac{1}{\sqrt{m}}(\Gamma^2_{\mathbf{g}_1,\xi_1}(\mathbf{w}, b), ..., \Gamma^2_{\mathbf{g}_m,\xi_m}(\mathbf{w}, b))^{\top} \tag{10}$$

Several strategies can be used to construct samples $(\xi_1, \mathbf{g}_1), ..., (\xi_m, \mathbf{g}_m)$, e.g. iid sampling or block-iid sampling with a fixed $\xi$ used within a block, but constructed independently for different blocks. In the experiments, we also choose: $\rho(\xi) = 2\pi i\xi$ and $\eta(\xi){=}1$.

**The case of discrete $\overline{\mathcal{P}}_j$ with finite number of atoms:** Assume that $(\xi^1, ..., \xi^K)$ is a sequence of atoms with the corresponding positive probabilities: $(p_1, ..., p_K)$. Then one can also construct $K$ pairs of RF-vectors $(\Phi^j(\mathbf{x}; k), \Psi^j(\mathbf{w}, b; k))_{k=1}^K$, each obtained by replacing $\overline{\mathcal{P}}_j$ with a distribution corresponding to a deterministic constant $p_k$ and get $\Phi^j(\mathbf{x}), \Psi^j(\mathbf{w}, b)$ by concatenating vectors from $(\Phi^j(\mathbf{x}; k))_{k=1}^K$ and $(\Psi^j(\mathbf{w}, b; k))_{k=1}^K$ respectively. This strategy is effective if $K$ is small.

Note that: $f(\mathbf{x}^{\top}\mathbf{w} + b) = \sum_{j=0}^3 \widehat{f_j}(\mathbf{x}, \mathbf{w}, b)$ and thus $\Phi_f(\mathbf{x})$ and $\Psi_f(\mathbf{w}, b)$ can be defined as:

$$\Phi_f(\mathbf{x}) = \mathrm{concat}\left((\Phi^j(\mathbf{x}))_{j=0}^3\right), \; \Psi_f(\mathbf{w}, b) = \mathrm{concat}\left((\Psi^j(\mathbf{w}, b))_{j=0}^3\right) \tag{11}$$

for the vector concatenation operation $\mathrm{concat}$, completing the description of the URF mechanism.

**Remark 3.1 (boundedness)** *We observe that for upper-bounded $\|\mathbf{x}\|_2, \|\mathbf{w}\|_2, |b|$, the entries of $\Phi_f(\mathbf{x})$ and $\Psi_f(\mathbf{w}, b)$ are also upper-bounded as long as $A < 0$. This follows directly from the formula for $\Lambda_{\mathbf{g}}(\mathbf{z})$ in Eq. 8.*

**Trigonometric activation functions:** Let us assume now that $f(z) = \sin(z)$ or $f(z) = \cos(z)$. Note that even though none of them has a Fourier Transform in the classical Riemannian sense, both have trivial Fourier Transforms in the broader distributional sense. To see that, we can rewrite both activations as: $\sin(z) = \frac{exp(iz) - \exp(-iz)}{2i}$ and $\cos(z) = \frac{exp(iz) + \exp(-iz)}{2}$. Therefore the corresponding distributions used in the URF derivations above become binary distributions over $\{-\frac{1}{2\pi}, \frac{1}{2\pi}\}$. This observation has interesting practical consequences, since it leads to the conceptually simple linearization of the FFLs applied in SIREN networks (see: Sec. 4.2).

## 3.2 BEYOND REGULAR FFLs: THE CURIOUS CASE OF THE ReLU-SNNK LAYER

We also propose another SNNK layer which is not directly inspired by any known FFL, but turns out to work very well in practice (see: Sec. 4.3.2). In this case, the mappings $\Phi$ and $\Psi$ are defined as: $\Phi(\mathbf{x}) = \mathrm{ReLU}(\frac{1}{\sqrt{l}}\mathbf{G}\mathbf{x}), \Psi(\mathbf{w}, b) = \mathrm{ReLU}(\frac{1}{\sqrt{l}}\mathbf{G}\mathbf{w})$ for the Gaussian matrix: $\mathbf{G} \in \mathbb{R}^{l\times d}$ with entries sampled independently at random from $\mathcal{N}(0, 1)$. One can ask a question what kernel does this pair of maps correspond to. It turns out that the answer is particularly elegant.

**Theorem 3.2 (arc-cosine kernels; Cho & Saul (2011))** *The $n$th-order arc-cosine kernel $K_n : \mathbb{R}^d \times \mathbb{R}^d \to \mathbb{R}$ is defined as: $K_n(\mathbf{x}, \mathbf{y}) = \frac{1}{\pi}\|\mathbf{x}\|_2^n\|\mathbf{y}\|_2^n J_n(\alpha_{\mathbf{x},\mathbf{y}})$, where $\alpha_{\mathbf{x},\mathbf{y}} \in [0, \pi]$ stands for an angle between $\mathbf{x}$ and $\mathbf{y}$ and $J(\theta) \stackrel{def}{=} (-1)^n(\sin(\theta))^{n+1}\frac{\partial^n}{\partial\theta^n}\left(\frac{\pi-\theta}{\sin(\theta)}\right)$. Then, $K_n$ can be linearized as: $K_n(\mathbf{x}, \mathbf{y}) = 2\mathbb{E}[\Gamma_n(\mathbf{x})^{\top}\Gamma_n(\mathbf{y})]$ for $\Gamma_n(\mathbf{v}) \stackrel{def}{=} \mathrm{ReLU}((\mathbf{v}^{\top}\boldsymbol{\omega})^n)$ and $\boldsymbol{\omega} \sim \mathcal{N}(0, \mathbf{I}_d)$.*

We conclude that our proposed ReLU-SNNK layer is a scalable version of the FFL defined as: $\mathbf{x}, \mathbf{W} \in \mathbb{R}^{l\times d}, \mathbf{b} \to (\frac{1}{2}K_1(\mathbf{w}^1, \mathbf{x}), ..., \frac{1}{2}K_1(\mathbf{w}^l, \mathbf{x}))^{\top}$.

**Remark 3.3** *The* ReLU-SNNK *layer is not a regular FFL since the values of its output dimensions cannot be re-written as* $f(\mathbf{x}^\top \mathbf{w}^i + b_i)$ *for some* $f : \mathbb{R} \to \mathbb{R}$ *(interestingly, after* $\Gamma$*-base pre-processing, it can be still interpreted as a dot-product kernel). This shows that the SSNK mechanism is capable of modeling relationships beyond those of regular FFLs.*

### 3.3 Bundling neural networks with SNNKs

We are ready to propose the neural network *bundling process*, relying on the SSNK-primitives. Consider the following deep NN module with input $\mathbf{x} = \mathbf{x}_0 \in \mathbb{R}^{d_0}$ and output $\mathbf{y} = \mathbf{x}_L \in \mathbb{R}^{d_L}$:

$$\begin{cases} \mathbf{x}_{i+1} = f_{i+1}(\mathbf{W}_i\mathbf{x}_i + \mathbf{b}_i); i = 0, ..., L-1, \\ \mathbf{x}_0 = \mathbf{x} \end{cases} \tag{12}$$

for: (1) matrices $\mathbf{W}_i \in \mathbb{R}^{d_{i+1} \times d_i}$, (2) bias vectors: $\mathbf{b}_i \in \mathbb{R}^{d_{i+1}}$, and (3) activations: $f_i : \mathbb{R} \to \mathbb{R}$.

To understand how the bundling process works, we start by replacing first FFL in Eq. 12 with its SNNK analogoue. We obtain the following computational block:

$$\begin{cases} \widehat{\mathbf{x}}_{i+1} = \widehat{f}_{i+1}(\widehat{\mathbf{W}}_i\widehat{\mathbf{x}}_i + \widehat{\mathbf{b}}_i) \text{ for } i = 0, ..., L-2, \\ \widehat{\mathbf{x}}_0 = \Phi_{f_1}(\mathbf{x}_0), \\ \widehat{\mathbf{W}}_0 = \mathbf{W}_1\Psi_{f_1}(\mathbf{W}_0, \mathbf{b}_0); \widehat{\mathbf{W}}_i = \mathbf{W}_{i+1} \text{ for } i = 1, ..., L-2, \\ \widehat{f}_{i+1} = f_{i+2}, \; \widehat{\mathbf{b}}_i = b_{i+1} \text{ for } i = 0, ..., L-2 \end{cases} \tag{13}$$

In the system of equations above, $\Psi_f(\mathbf{W}_0, \mathbf{b}_0)$ is a matrix with transposed rows of the form: $\Psi_f(\mathbf{W}_0^j, \mathbf{b}_0^j)$, where $\mathbf{W}_0^j$ for $j = 0, ..., d_1 - 1$ are the transposed rows of $\mathbf{W}_0$ and $\mathbf{b}_0 = (\mathbf{b}_0^0, ..., \mathbf{b}_0^{d_1-1})^\top$. We have thus successfully replaced a module of $L$ feedforward layers with a module of $(L-1)$ feedforward layers. By continuing this procedure, we can ultimately get rid of all the FFLs and obtain an estimator $\overline{\mathbf{y}}$ of $\mathbf{y}$, given as: $\overline{\mathbf{y}} = \overline{\mathbf{W}}\overline{\mathbf{x}}$, where

$$\begin{cases} \overline{\mathbf{x}} = \Phi_{f_L}\left(\Phi_{f_{L-1}}(...\Phi_{f_1}(\mathbf{x}_0)...)\right) \\ \overline{\mathbf{W}} = \Psi_{f_L}\left(\mathbf{W}_{L-1}\Psi_{f_{L-1}}(...\mathbf{W}_2\Psi_{f_2}(\mathbf{W}_1\Psi_{f_1}(\mathbf{W}_0, \mathbf{b}_0), \mathbf{b}_1)..., ), \mathbf{b}_L)\right) \in \mathbb{R}^{d_L \times m} \end{cases} \tag{14}$$

This has several important consequences. In inference, replacing matrices $\mathbf{W}_0, ..., \mathbf{W}_{L-1}$ with one matrix $\overline{\mathbf{W}}$ is a effectively a compression scheme (that does not necessarily need to be applied to all the layers, but a particular consecutive set of layers of interest). If we apply bundling process to the entire deep neural network, we effectively provide its two-tower factorization with input disentangled from the parameters. In training, we can treat $\overline{\mathbf{W}}$ as an unstructured parameter matrix and directly learn it (see results in Appendix J.3, table 5). Since the output $\overline{\mathbf{y}}$ is now modeled as an action of the unstructured learnable matrix $\overline{\mathbf{W}}$ on the *pre-processed* input $\overline{\mathbf{x}}$, for several loss functions there exists an explicit formula for the optimal $\overline{\mathbf{W}}$. This is the case in particular for the standard regression loss (see discussion in Appendix J.3). If bundling is applied to a particular module, backpropagation through it is not necessary since there exists an explicit formula for the corresponding Jacobian.

## 4 Experiments

We present an extensive empirical evaluation on SNNK on a wide range of experiments. More details on each of the experiments can be found in the Appendix G.

### 4.1 Pointwise Kernel Estimation

As a warm-up, we test the accuracy of the applied RF-mechanisms on synthetic data. We take $d = 2000$ and $l = 1$. We consider: **(a)** a SIREN-FFL with the activation function $f(u) = \sin(u)$ and bias $b = 0.5$, **(b)** an arc-cosine-FFL from Sec. 3.2. The entries of the weight vectors $\mathbf{w}$ and the inputs to the layers are taken independently from $\frac{1}{\sqrt{d}}\text{Unif}(0, 1)$. We report the mean relative error of the NN output (by averaging over $s = 500$ instantiations of the RF-mechanism) made by the RF-based estimator as well as the empirical standard deviation as a function of the number of random projections. This setup corresponds to quantifying the accuracy of the kernel estimator pointwise. The results are presented in Fig. 2 (g and h). Our SNNK provided an accurate approximation with a much smaller number of random projections than the dimensionality $d$ of the input vectors.

### 4.2 Toy Experiments

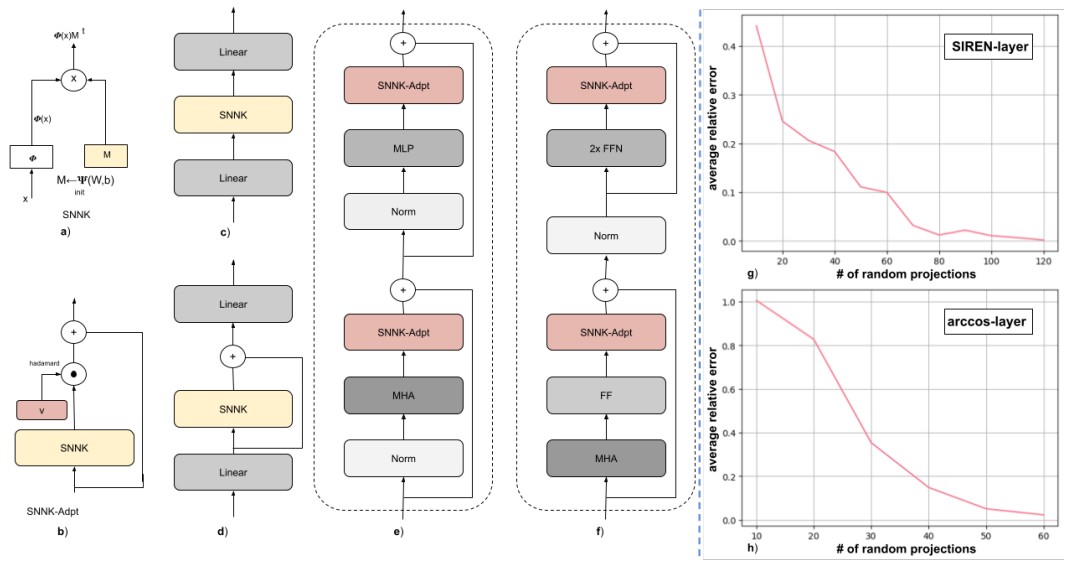

Figure 2: Architecture for **(a)** SNNK layer (see Section C), **(b)** SNNK-Adpt layer **(c)** image fitting (SIREN), MNIST and UCI experiments, **(d)** SNNK-QPNN model, **(e)** SNNK-inspired Adapter-ViT layer, **(f)** SNNK-inspired Adapter-BERT layer. **(g,h):** The relative error (obtained by averaging over $s = 500$ instantiations of the RF-mechanism) made by the RF-based estimator on the particular entry of the output of the: **(g)** SIREN-FFL and **(h)** arc-cosine-FFL as a function of the number of random projections $p$ (see: Sec. 4.1). The maximum $p$ for (g) is larger than for (h), as (g) in theory produces larger variance per random projection. The corresponding standard deviations are negligible: **(g)** $5 \cdot 10^{-8}, 10^{-12}, 5 \cdot 10^{-8}, 10^{-8}, 10^{-12}, 2.5 \cdot 10^{-9}, 10^{-12}, 5 \cdot 10^{-9}, 10^{-12}, 10^{-12}, 10^{-10}, 10^{-12}$, **(h)** $10^{-12}, 3 \cdot 10^{-8}, 3 \cdot 10^{-8}, 2 \cdot 10^{-8}, 10^{-12}, 5 \cdot 10^{-9}$.

SNNKs are versatile and can be used as a drop-in replacement for FFLs in a wide variety of NNs like the SIREN network Sitzmann et al. (2020), QPNN - a Physics-inspired Neural Network (PINN) to solve the Hamiltonian for quantum physical systems (Sehanobish et al., 2021) and a simple multi-layer perceptron (MLP) for classification on MNIST (LeCun & Cortes, 2010). We use the sine activation variant for the first two experiments and the ReLU variant for MNIST. 32 random features are used for the solution of the 2-body problem and MNIST and 64 random features for the image fitting problem. We match the performance of the baseline NNs on the 2-body

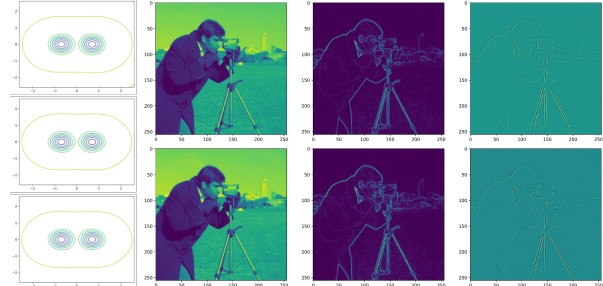

Figure 3: (1) **Left column** : Injecting SNNK in a PINN network to approximate the potential energy of the 2-body sytem. Top to bottom : Ground truth potential, Learned potential by QPNN (Sehanobish et al., 2021) and QPNN-SNNK. QPNN-SNNK can learn the potential function perfectly even using less trainable parameters than the baseline QPNN. (2) **Rightmost three column** : Siren network on the first row, fitting not only the image, but also the gradients. SNNK on the bottom row produces an accurate approximation of the above.

and the image fitting problem (see figure 3) and outperform the baseline on MNIST (Figure 9), while incurring lower training costs. For additional details regarding these experiments, see Appendix G.1.

## 4.3 FINETUNING EXPERIMENTS

In this subsection, we show how SNNKs can be used for parameter efficient finetuning. For NLP experiments, we use the GLUE benchmark consisting of 8 different natural language understanding tasks (Wang et al., 2018). For vision tasks, we use CiFAR-10, CiFAR-100 (Krizhevsky et al., 2009) and ImageNet-1k (Deng et al., 2009). BERT-base (Devlin et al., 2019) is used as a backbone for text experiments and ViT (Kolesnikov et al., 2021) for image experiments. Our code is built on top

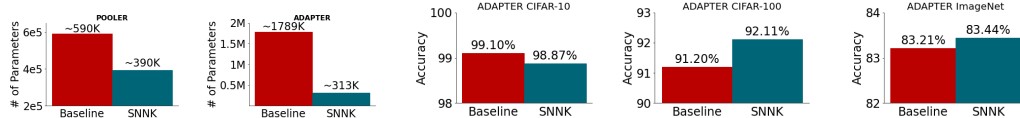

Figure 4: Comparison of trainable parameters between various layers/modules and the drop in replacement NNK layers. Results for CiFar-10, CiFar-100 and ImageNet are for SNNK-Adapter models.

of Transformers (Wolf et al., 2020) and adapter Transformer library (Pfeiffer et al., 2020). Detailed comparisons with various baselines can be found in Appendix K and additional experiments in Appendix J.

### 4.3.1 LINEARIZING THE POOLER LAYER IN TRANSFORMERS

For text classification tasks, a SNNK layer can be used as a drop-in replacement for the pooler layer which is a linear layer with a tanh activation. For these set of experiments, the base model is frozen and only the pooler and the classifier weights are tuned. We get computational gains as the number of random features employed by SNNK is smaller than that of the hidden size of the Transformers. More details are presented in Appendix G.2.

On GLUE dev set, our SNNK-linearized pooler models outperform the baselines on 5 out of 8 tasks (Table 1 (top half)). Additional results can be found in Appendix J.

In this setting, the linearized pooler weights can be merged with the classifier weights to create a weight matrix of size (# number of random features $\times$ number of classes) and then one can simply store the newly merged layer instead of separately storing the trained classifier and pooler layers. This dramatically *reduces* the storage from 18.92 Megabit to only .**02** Megabit leading to a compression factor of $\mathbf{1/1000}$. More details are presented in Appendix E. Ablation studies on the number of random parameters for this experimental setting are presented in Appendix I.

Table 1: SNNK experiments on GLUE benchmarks. MCC score is reported for CoLA, F1 score is reported for MRPC and QQP, Spearman correlation is reported for STSB. Accuracy scores are reported for the other tasks. All results are obtained by averaging over 5 seeds.

| Dataset | RTE | MRPC | QNLI | QQP | SST-2 | MNLI | STSB | COLA |
|---|---|---|---|---|---|---|---|---|
| Bert-baseline (Lee et al., 2019) | 57.5 | 81.5 | **74.5** | **72.0** | 84.9 | **56.4** | 78.1 | 29.4 |
| Cosine-SNNK-pooler (ours) | **61.36** $\pm$ 1.15 | **82.07** $\pm$ 1.07 | 73.5 $\pm$ 0.22 | 70.43 $\pm$ 0.17 | **85.21** $\pm$ 0.34 | 52.69 $\pm$ 0.32 | **78.93** $\pm$ 0.37 | **35.81** $\pm$ 0.96 |
| Adapter-baseline (Moosavi et al., 2022) | 63.83 $\pm$ 1.4 | 84.8 $\pm$ 1.07 | **90.63** $\pm$ 0.26 | **88.12** $\pm$ 0.14 | 91.74 $\pm$ 0.36 | **83.53** $\pm$ 0.19 | 88.48 $\pm$ 0.14 | 56.51 $\pm$ 0.84 |
| *AA* (Moosavi et al., 2022) | 64.25 $\pm$ 1.72 | 85.09 $\pm$ 1.06 | 89.96 $\pm$ 0.25 | 88.09 $\pm$ 0.16 | 91.31 $\pm$ 0.51 | 82.89 $\pm$ 0.43 | 88.25 $\pm$ 0.17 | 51.44 $\pm$ 1.82 |
| ReLU-SNNK-Adapter (ours) | **69.68** $\pm$ 1.24 | **91.26** $\pm$ 1.39 | 90.44 $\pm$ 0.16 | 85.82 $\pm$ 0.23 | **92.31** $\pm$ 0.27 | 82.06 $\pm$ 0.17 | **88.81** $\pm$ 0.14 | **58.21** $\pm$ 0.63 |

### 4.3.2 SNNK-INSPIRED ADAPTER LAYERS

Adapters in Transformers were first introduced in (Houlsby et al., 2019) and there has been a lot of work designing different architectures (Pfeiffer et al., 2020; Karimi Mahabadi et al., 2021; Moosavi et al., 2022) and unifying various paradigms (Moosavi et al., 2022; He et al., 2022a). Adapters are bottleneck MLPs which are (generally) added twice to each Transformer layer. In our work, we replace each adapter block by a single SNNK layer (Figure 2 (e) and (f)) using only **16** random features resulting in a big drop of training parameters (see Figure 4). Figure 4 (b) shows the architecture of SNNK-inspired adapter layers. Additional details are presented in Appendix D.

As is customary for adapter experiments, base model is frozen and only the adapters and classifier are tuned. Table 1 (bottom half) shows our results on using SNNK layers in place of adapters on the GLUE dev set. We outperform the baseline on 5 out of 8 datasets while employing only $\mathbf{1/3}$ of the training parameters. On MNLI, it is noted in (Houlsby et al., 2019), that using smaller adapter size causes worse performance and performance boost can be achieved by increasing the size of the adapter (256 is used in their case). Similar to this observation, we note that we can improve performance and match the baselines on large datasets (ex. MNLI, QNLI) as we increase the number of random features (see Figure 5). Our method also produces competitive performance on image datasets including Cifar-10, Cifar-100 and ImageNet.(see Figure 4 (right 3 figures)). Detailed comparisons with SOTA parameter efficient finetuning methods can be found in Table 7 (vision tasks) and in Table 8 (GLUE tasks).

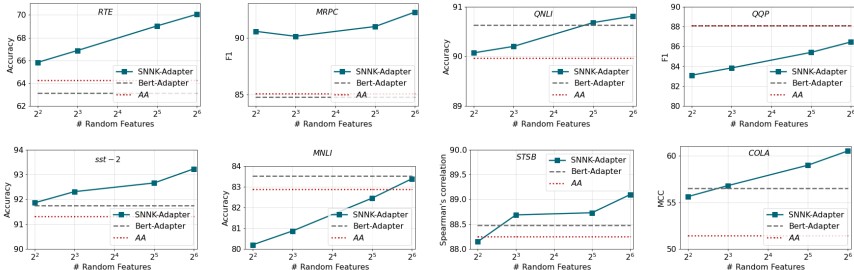

Figure 5: Ablation with different number of random features for the ReLU-SNNK-adapter experiments on the GLUE dev set. $AA$ is the reported adaptable adapter numbers in Moosavi et al. (2022).

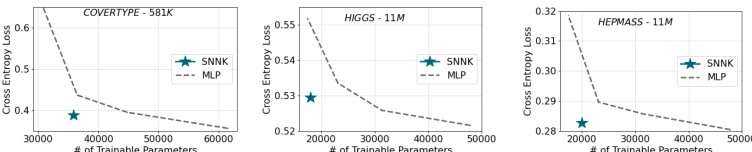

Figure 6: Comparison of CE loss for SNNK vs different sizes of MLP on UCI datasets.

Moreover, we note that our methods are completely orthogonal to techniques such as gating mechanism in (Mao et al., 2022) or algorithms relying on dropping suitable adapter layers (Moosavi et al., 2022; Rücklé et al., 2021). Thus it can be easily combined with them.

## 4.4 UPTRAINING TRANSFORMERS

In this section, we report results of replacing part of the Feed-Forward Network (FFN) block in Transformers with SNNKs. Details are in the Appendix J.4 for brevity, here we summarize our findings. We observe from Table 6 that replacing FFL blocks with SNNK layers reduces the number of parameters and FLOPS for both training and inference. This followed by our bundling process leads to a large reduction in size and inference time of the bundle. For example, for BERT and ViT models, replacing top-6 Transformer layer's MLP block with SNNK reduces the size of the model from 440 Mb to 226.71, and 346 Mb to 176.42 Mb respectively. Figures 12 and 13 demonstrate that reducing the model size and inference speed by 40-50% has minimal impact on accuracy for both NLP and Image classification tasks.

## 4.5 EXPERIMENTS ON UCI DATASETS

We have conducted experiments with a variety of real-world datasets found in the UCI Machine Learning Repository (UCI MLR).[1] We trained a three-layer MLP model as baseline (see Appendix Sec. H.5 for details). We varied the output of the middle-layer to train MLPs with different sizes. For our method, we replace the middle-layer with SNNK (Figure 2 (c)). SNNK matches or outperforms the baseline while using only a fraction of the training parameters (Figure 6).

## 5 CONCLUSION

We present scalable neural network kernels (SNNK), a novel efficient NN computational model that can be used to replace regular feedforwards layers in MLPs, where inputs and parameters are disentangled and connected only in the final computation via a dot-product kernel. We introduce a general mechanism of the universal random features (URFs) to instantiate SNNKs, show that SNNKs are capable of encoding subtle relationships between parameter- and input-vector beyond functions of their dot-products and finally, explain how they lead to the compactification of the NN stack via the so-called bundling process. We complement our theoretical findings with the exhaustive empirical analysis, from pointwise kernel estimation to training Transformers with adapters.

---

[1]https://archive.ics.uci.edu/ml/index.html

ETHICS STATEMENT

This paper focuses mostly on the algorithmic properties of the techniques linearizing kernels associated with feedfoward layers' calculations for the computational gains. The experiments with adapter-based fine-tuning of Transfomers are presented to illustrate the main concepts. It should be noted though that Transformers should be used cautiously given their considerable computational footprint (improved though while adapters are applied) and the corresponding carbon footprint.

AUTHOR CONTRIBUTIONS

AS and KC led the project. AS ran several empirical studies on GLUE, CiFar-10 and CiFar-100 datasets and proposed several strategies for efficiently using SNNK-layers within Transformer models. KC proposed FFL linearization-schemes, URFs, the bundling mechanism and implemented all linearization-schemes. YZ ran empirical studies on GLUE, CiFar-10 and CiFar-100 datasets. AD implemented and ran all UCI experiments, helped with GLUE/image experiments, proposed strategy for efficiently using SNNK-layers and created all figures in experiments. VL proposed an idea to linearize FFLs by disentangling inputs from weights. All authors contributed to the writing of the manuscript.

REPRODUCIBILITY STATEMENT

Hyperparameters to reproduce each experiment is detailed in section H. The code is provided at `https://github.com/arijitthegame/neural-network-kernels`.

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

## A  TOWARDS URFs: ADDITIONAL INTUITION

In this section, we provide an additional intuition that led us to the mechanism of URFs. Our first observation is that if one has a mechanism for the linearization of the softmax kernel $\mathrm{K}(\mathbf{x}, \mathbf{y}) = \exp(\mathbf{x}^\top \mathbf{y})$, i.e. a randomized mapping $\phi : \mathbb{R}^d \to \mathbb{R}^m$ such that: $\mathrm{K}(\mathbf{x}, \mathbf{y}) = \mathbb{E}[\phi(\mathbf{x})^\top \phi(\mathbf{y})]$, then this mechanism automatically provides an approximate linearization of the kernel defined as:

$$\mathrm{K}(\mathbf{x}, \mathbf{y}) = \sum_{i=1}^{l} c_i \exp(s_i \mathbf{x}^\top \mathbf{y}), \tag{15}$$

for some coefficients: $c_1, ..., c_l, s_1, ..., s_l$. To see that, note that one can simply define the random feature map $\Psi$ for that kernel as:

$$\Psi(\mathbf{z}) = \mathrm{concat}\left(\sqrt{c_1}\phi(\sqrt{s_1}\mathbf{z}), ..., \sqrt{c_l}\phi(\sqrt{s_l}\mathbf{z})\right), \tag{16}$$

where $\mathrm{concat}$ stands for the concatenation operator. Alternatively, if in addition $c_1, ..., c_l > 0$, one can first sample the index $k \in \{1, ..., l\}$ from the discrete distribution $(p_i, ..., p_l)$ with $p_t = \frac{c_t}{\sum_{n=1}^{l} c_n}$ and then define $\Psi$ as:

$$\Psi_k(\mathbf{z}) = \sqrt{\sum_{n=1}^{l} c_n \phi(\sqrt{s_k}\mathbf{z})} \tag{17}$$

The mechanism of URFs is the natural extension of this observation to the setting, where the kernel cannot be given by the sum from Eq. 15, but the formula with the sum replaced by the integral (which immediately leads to the representation of the function as the Fourier Transform of its inverse Fourier Transform or vice versa: inverse Fourier Transform of its Fourier Transform). The only remaining step is to choose a right mapping $\phi$ for the estimation of the regular softmax kernel and here we have decided to leverage the recently introduced improvement of the positive random feature map mechanism from Likhosherstov et al. (2022).

## B  PROPAGATION OF THE ERROR OVER THE NETWORK

In this section, we show how the error accumulates when we try to bundle a deep feedforward network.

The variance of the estimation of the kernel value $\mathrm{K}(\mathbf{x}, \mathbf{y})$ is proportional to $\frac{1}{m}$, where $m$ is the number of random features. This is the case since its estimator can be rewritten as:

$$X = \frac{1}{m} \sum_{i=1}^{m} X_i,$$

where each $X_i$ provides an unbiased estimation of the kernel value. Since random variables $X_i$ are independent, using Azuma's Inequality, we can also conclude that if $|X_i| \leq c$, then:

$$P[|X - \mathrm{K}(\mathbf{x}, \mathbf{y})| \geq \epsilon] \leq 2 \exp\left(-\frac{\epsilon^2}{8mc^2}\right), \tag{18}$$

for any $\epsilon > 0$. Recall from $\exp(\widehat{\mathbf{x}}^\top(\xi)\widehat{\mathbf{w}}(\xi)) = \mathbb{E}_{\mathbf{g} \sim \mathcal{N}(0, \mathbf{I}_d)}[\Lambda_\mathbf{g}(\widehat{\mathbf{x}})\Lambda_\mathbf{g}(\widehat{\mathbf{w}})]$, where $\Lambda_\mathbf{g} : \mathbb{R}^d \to \mathbb{R}$ is defined as follows for $A \leq 0$:

$$\Lambda_\mathbf{g}(\mathbf{z}) = (1 - 4A)^{\frac{d}{4}} \exp(A\|\mathbf{g}\|_2^2 + \sqrt{1 - 4A}\mathbf{g}^\top \mathbf{z} - \frac{\|\mathbf{z}\|_2^2}{2}) \tag{19}$$

Note that the boundedness condition holds if $A < 0$ (see Remark 3.1).

Now assume that the kernel function K satisfies (in the region of interest): $|\mathrm{K}(\mathbf{x}, \mathbf{w}) - \mathrm{K}(\mathbf{u}, \mathbf{w})| \leq \delta(a)$, as long as $\|\mathbf{x} - \mathbf{u}\|_1 \leq a$ for some function $\delta$. Note that this is not a strong assumption as any continuous function on a compact subset of $\mathbb{R}^n$ is uniformly continuous, i.e. satisfies the above condition for some $\delta$, where $\delta(a) \to 0$ as $a \to 0$.

We then conclude that the probability that the approximate output of the bundled $d$-layer neural network differs from the exact output by at least $\epsilon + \delta(\epsilon) + \cdots + \delta(\cdots \delta(\delta(\epsilon)))$ (composition of $(d - 1)$ $\delta$-functions) in the $L^1$-norm is upper-bounded by the RHS from the Inequality 18, but with an extra multiplicative factor $d$ (coming from the union-bound). We see that the number of random features needed for an accurate approximation is larger if K grows faster (e.g. is $L$-Lipschitz with larger constant $L$).

## C  SNNK Layers and Relation to Linearization of 2-layer Neural Networks

In this subsection, we provide details on the training of the SNNK layers and how they can be injected in place of MLP layers. We use the PyTorch style of Linear layer in our notations, namely right multiplication by $\boldsymbol{W}^\top$. Recall an MLP layer with weight matrix $\boldsymbol{W}$, a bias $\boldsymbol{b}$ and a non-linear function $\sigma$ takes an input $\boldsymbol{X}$ and computes

$$f(\boldsymbol{X}) = \sigma(\boldsymbol{X}\boldsymbol{W}^\top + \boldsymbol{b}) \sim \Phi(\boldsymbol{X})\Psi(\boldsymbol{W}, \boldsymbol{b})^\top \qquad (20)$$

Rewriting $\boldsymbol{A}$ as $\Psi(\boldsymbol{W}, \boldsymbol{b})$, one can think of $\boldsymbol{W}$ as fixed weights while $\boldsymbol{A}$ are trainable which is exactly equation 3. Moreover $\boldsymbol{A}$ has fewer parameters than $\boldsymbol{W}$, resulting in parameter efficient training.

Using this above intuition, one can seamlessly plug SNNK in place of either pretrained network where the input to $\Psi$ will be pretrained weights or random weights in case of untrained networks. We do not train $\boldsymbol{W}$ in this case, but only $\boldsymbol{A}$.

## D  SNNK-inspired Adapter Blocks

We think of adapters (discarding the nonlinearity) as a low rank factorization of a linear layer. If the weight matrix $\boldsymbol{W}$ of the linear layer is of size $(d \times d)$ and $k$ is the low rank of the factorized matrices where $k << d$, the number of trainable parameters is $2 \times k \times d$ (again discarding the bias terms) as opposed to $d \times d$. However, we consider the problem of linearizing the entire block by one feature matrix of size $(d \times \text{random features})$ and we similarly transform the input tensors and compute the matrix multiplication in the feature space. In this setting, we ignore the bias term $\boldsymbol{b}$ as $\boldsymbol{b}$ is initialized as the zero vector and so it does not change the initial feature matrix. Thus our implementation of linearized adapter looks like :

$$\boldsymbol{Y} := \text{SNNK}(\boldsymbol{X}) = \Phi(\boldsymbol{X})\Psi(\boldsymbol{W})^\top \qquad (21)$$

where $\Phi$ and $\Psi$ are suitably chosen feature maps. As $\boldsymbol{W}$ is randomly initialized, we can treat $\Psi(\boldsymbol{W})$ as a random unstructured matrix $\boldsymbol{A}$ and update $\boldsymbol{A}$. We use a residual connection as in (Houlsby et al., 2019). It is well-known (Houlsby et al., 2019; He et al., 2022a; Pfeiffer et al., 2020) that for stable training the adapter block should behave like the identity matrix at initialization. We introduce a vector $\boldsymbol{v}$ that we call the gating vector or the modulating vector that is initialized as the zero vector. Thus the equations of our adapter block becomes :

$$\boldsymbol{Y} = (\boldsymbol{v} \odot \text{SNNK}(\boldsymbol{X})) + \boldsymbol{X} \qquad (22)$$

$\odot$ stands for Hadamard or element-wise product.

At this point, we would like to give some motivation regarding the use of this vector and discuss initialization schemes. One simple way to initialize the entire block as the identity, is to choose $\boldsymbol{A}$ as the $\boldsymbol{0}$ matrix but that in turn leads to optimization difficulties. Another initialization scheme would be to initialize $\boldsymbol{A}$ from a Gaussian centered around $0$ with small variance, but does not lead to good performance. Other detailed initialization schemes are not studied and are beyond the scope of this work. Meanwhile, adding the gating vector allows us to initialize the block as the identity matrix, leading to stable training.

Thus the number of training parameters in this regime is $(d+1) \times \text{random features}$ which is considerably lower than the adapters if the number of random features are small. For all our experiments, we only use **16** random features resulting in lowering the number of training parameters compared to the baseline adapters. Our SNNK-inspired adapters can be used in different configurations and can be also combined with SOTA adapter-based methods and we leave that to future work.

## E  Bundling the Pooler and the Classifier Layers

The *bundling* of the pooler and the classifier layer takes a particularly simple form in this case: If $\boldsymbol{W}_p, \boldsymbol{b}_p$ (resp. $\boldsymbol{W}_c, \boldsymbol{b}_c$) are the weight matrices and biases of the pooler and classifier layer resp and $\sigma$ be the tanh activation function. Then :

$$\sigma(\boldsymbol{X}\boldsymbol{W}_p^\top + \boldsymbol{b}_p)\boldsymbol{W}_c^\top + \boldsymbol{b}_c \sim \Phi(\boldsymbol{X})\Psi(\boldsymbol{W}_p, \boldsymbol{b}_p)^\top \boldsymbol{W}_c^\top + \boldsymbol{b}_c \qquad (23)$$

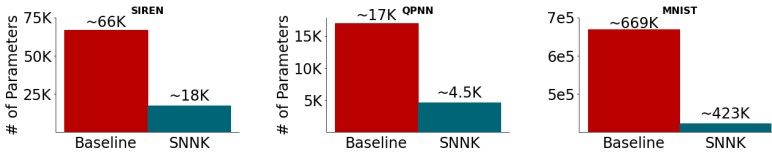

Figure 7: Comparison of trainable parameters between the baseline models for toy experiments and the SNNK-counterparts.

$\Psi(\boldsymbol{W}_p, \boldsymbol{b}_p)$ is a matrix of size $d \times k$, where $d$ is the dimension of the transformer, $k$ is the number of random features and $k < d$. $\boldsymbol{W}_c$ is a matrix of size $d \times c$, where $c$ is the number of output classes. Thus, instead of storing the two matrices, we can only store the product of the 2 matrices which is of size $k \times c$, resulting in huge storage savings. Moreover during deploying, one can only use the smaller matrix as the output layer resulting in lower flops.

## F   DATASETS

We describe the dataset statistics used in different experiments. Glue is a benchmark dataset in NLP comprising of 8 different natural language understanding (NLU) tasks (Wang et al., 2018). Table 2 shows the train, dev splits for various tasks. CiFar-10 consists of 50k natural images for training split

Table 2: Statistics of Glue datasets

| Dataset | RTE | MRPC | QNLI | QQP | SST-2 | MNLI | STSB | COLA |
|---|---|---|---|---|---|---|---|---|
| Train/Dev | 2.49k/277 | 3.67k/408 | 105k/5.46k | 364k/40.4k | 67.3k/872 | 393k/9.83k | 5.75k/1.5k | 8.55k/1.04k |

into 10 categories with 5k images in each category and 10k images for testing. CiFar-100 consists of 50k natural images for training split into 100 categories with 500 images in each category and 10k images for testing. MNIST is a dataset of handwritten digits, consisting of 60k training examples across 10 categories and 10k test examples. For these datasets, we use a 25% stratified random sampling from the training set to create a validation set which is used to select the best model to run on the holdout set.

We use three large UCI classification datasets CoverType[2] ( $510K$ points, $dim = 54$), HIGGS[3] ( $11M$ points, $dim = 28$) and HEPMASS[4] ( $11M$ points, $dim = 28$) to evaluate SNNK.

## G   EXPERIMENTS

In this section, we provide additional details for various experiments.

### G.1   ADDITIONAL DETAILS ON TOY EXPERIMENTS

In the following subsections, we provide additional details on the toy experiments. The aim for conducting the toy experiments is to showcase the versality of the SNNK-layer that can be used as a drop-in replacement for MLP layers in different optimization problems.

### G.1.1   QPNN

QPNN is a simple PINN-styled 3 layer neural network (Sehanobish et al., 2021). It learns the potential energy of a quantum system in an unsupervised manner by learning the principle of conservation of energy (using Schrodinger's equation). The hidden layer of the QPNN is of size $128 \times 128$ and in this work, we replace the hidden layer by our SNNK layer with 32 random features, resulting

---

[2]https://archive.ics.uci.edu/dataset/31/covertype
[3]https://archive.ics.uci.edu/dataset/280/higgs
[4]https://archive.ics.uci.edu/dataset/347/hepmass

in efficient computation of the potential energy of the 2-body system. We use the cosine variant of SNNK for generate the plots in the main paper. We observe similar performance for the sine variant of SNNK.

### G.1.2 SIREN

Siren network is a MLP with sine activations introduced in (Sitzmann et al., 2020). Siren network for the image fitting experiment is a 3 layer neural network with hidden dimension matrices of sizes $256 \times 256$. As is shown in the original work, Siren network can not fit the image but can also represent the derivatives of the signal. We replace the hidden layer with our sine variant of the SNNK layer with $64$ random features, resulting in modest computational gains. We show that like the original Siren network, our SNNK-augmented Siren network can also accurately model the derivatives of the signal.

The cosine variant of SNNK performs similarly in this task. This is not surprising since the derivative of a siren network is also a siren network, and cosine function is a shifted sine function and so they behave similarly.

### G.1.3 CLASSIFICATION ON MNIST

The baseline NN is a simple 3-layer feedforward layer with hidden dimension to size $512 \times 512$ with ReLU activation. Like our previous experiments, we replace the hidden layer by our ReLU-variant of the SNNK layer. We train the baseline as well as SNNK-MLP with various number of random features for 25 epochs and measure their accuracy on the test set (see figure 9 (left)).

Moreover, we observe lower training losses and stable training for SNNK-MLP across different number of random features (see figure 9 (middle)) .

### G.2 EXPERIMENTS ON LINEARIZING THE POOLER LAYER IN PRETRAINED TRANSFORMERS

Many encoder only pretrained transformers use a special token called [CLS] that is appended in front of the input sequence which can be used for classification tasks. After the transformer blocks, these encoder models employ a pooling layer on the final hidden state of the [CLS] token as the final representation for the sequence which is then passed to a classifier layer for a classification or regression task. The pooling layer is a linear layer of size $d \times d$, where $d$ is the hidden dimension of the transformer with a tanh activation. For the baseline results, the BERT layers are frozen and only the classifier and pooler is tuned.

We found the SNNK-variant with the cosine variant to be particularly performant in various scenarios so we use it as a proxy. Both the cosine and the tanh functions have the same range of values so we find no instabilities or difficulties in training. For these experiments, we replace the pooling layer by our NNK layer employing fewer random features than the hidden dimension. Detailed discussion about the bundling procedure is explained in Appendix E. We would like to note that in image classification tasks, the pooler layer is not used, only the final [CLS] token is used for classification.

### G.3 EXPERIMENTS ON SNNK-INSPIRED ADAPTERS

Adapters in transformers were first introduced for NLP tasks in (Houlsby et al., 2019) and there has been a lot of work designing different architectures (Pfeiffer et al., 2020; Karimi Mahabadi et al., 2021; Moosavi et al., 2022) and unifying various paradigms (Moosavi et al., 2022; He et al., 2022a). Adapters are bottleneck MLPs which are added twice to each Transformer layer, one after the attention layer and once after the feedforward layer. For simplicity, we choose to work with the original Houlsby configuration in this work.

Following the success of adapters for NLP tasks, several authors have proposed various variants of adapters in ViT (Chen et al., 2022; He et al., 2022b; Yu et al., 2022). Like the BERT-adapter, we use the adapter implementation for ViT-adapters using the Houlsby configuration as described in (Pfeiffer et al., 2020).

## H    Hyperparameters

In this section, detailed hyperparameters for each experimental setting is presented. All experiments on the smaller dataset are run on a Google Colab free version (T4 GPU), while experiments on the larger datasets used V100 GPU and 40Gb A100 GPUs in Google Colab. For the GLUE experiments, we use the sequence length of $128$. For experiments on image datasets, we use the ViT checkpoint *'google/vit-base-patch16-224-in21k'* which is a ViT-base model pretrained on ImageNet-21k (Deng et al., 2009). We use $224 \times 224$ resolution for the images for all the experiments. For experiments on text datasets, we use the bert-base-uncased checkpoint.

### H.1    Adapter Finetuning

For these experiments, we use a learning rate of $1e - 3$, batch size of 64 and a constant scheduler with warmup steps $6\%$ of the total number of training steps with AdamW optimizer (Loshchilov & Hutter, 2019). All models converge within 5-30 epochs depending on the size of the dataset. For the baseline experiments, we choose an adapter size of $48$ for all experiments. For our SNNK-adapter experiments, we merely use the hyperparameters found in (Pfeiffer et al., 2020). We realize that such a choice of hyperparameter may not be suitable for optimal performance but detailed hyperparameter tuning is beyond the scope of the work.

For experiments on image datasets we also use gradient clipping to clip the gradients to global norm 1. We set number of random features as 16 for all adapter finetuning experiments except ImageNet for which we used 32.

### H.2    Pooler Finetuning

For these experiments, we use learning rate of $1e - 3$, batch size of 64 and a weight decay of $1e - 4$ with AdamW optimizer. All models converge within 5-20 epochs depending on the size of the dataset.

For the image experiments, we also use a constant scheduler with warmup steps $1\%$ of the total number of training steps. We also divide the inputs to our SNNK layer as well the initial weights by $d^{.25}$, where $d$ is the hidden dimension of the Transformer. Note that this is the same transformation that is used in (Choromanski et al., 2021).

### H.3    Experiments on Bundled Networks

For these experiments, we do a hyperparameter search over learning rates in {1e-2, 5e-3, 2e-3, 1e-3 }. We use a batch size of $64$, Adam optimizer and do not employ any regularization techniques.

### H.4    Toy Experiments

For the experiments on image fitting and computing the potential energy of the 2-body system, we use the default parameters as used by the authors of the original papers. For MNIST experiment, we use a batch size of 32, learning rate of .001 with Adam optimizer. We also use a dropout of .2 after every layer except the output layer.

### H.5    UCI Experimental Details

A three layer MLP is used as a baseline. We use dropout and ReLU activation in the first two layers, with the layers being : 1) $\text{ReLu}(\text{Linear}(\text{dim}, 512))$ and 2) $\text{ReLu}(\text{Linear}(512, d_{\text{param}}))$ and the output layer being : $\text{output} = \text{Linear}(d_{\text{param}}, numClasses)$, where dim is the dimension of the input (data dependent), $d_{\text{param}}$ is a hyperparameter that we vary to get all the relevant plots and $numClasses$ is the total number of classes to be predicted. For our method, we replace the second layer with a SNNK layer.

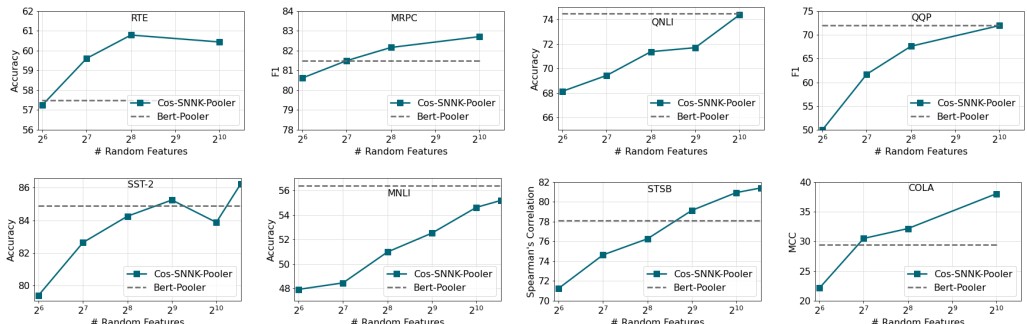

Figure 8: Ablation with different number of random features for the Cosine-SNNK-pooler experiments on the GLUE dev set. $Bert - Pooler$ baseline is from Lee et al. (2019).

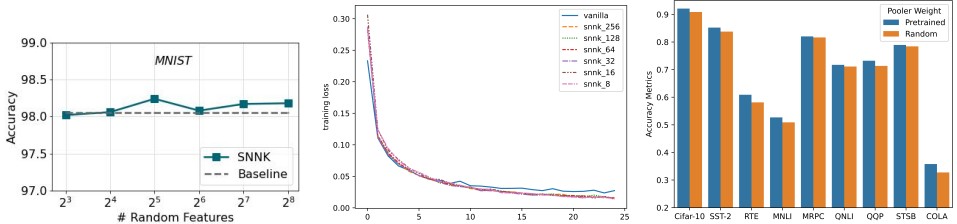

Figure 9: Ablation results for MNIST dataset. **Left :** Accuracies of SNNK-MLP for different number of random features. **Middle :** Training losses for the corresponding models on MNIST training set. **Right :** Starting with pretrained Pooler weights produce better results than randomly initialized layer for both image and text datasets.

## H.6 UPTRAINING EXPERIMENTS

We use 8 random features, AdamW optimizer and linear learning schedule with warmup of 6% of the total optimization steps for all uptraining experiments. For the larger GLUE datasets (SST-2, MNLI, QNLI, QQP) we use a weight decay of .0001 and for the smaller datasets we use a weight decay of .1 and a gradient clipping of 1. For image datasets, we use a weight decay of .001. We train for 5-20 epochs depending on the dataset, and selecting the best model based on validation loss.

## I   ABLATION STUDIES

In this section, we present additional ablations in different experimental settings. Figure 9 (Left and Middle) shows accuracy on MNIST test set and the training curves for SNNK-MLP for different number of random features.

We also show the importance of starting with the pretrained pooler weights over a randomly initialized layer. Our results show that the initial features coming from the pretrained pooler weights produce better results on a wide range of text and image datasets.

Finally we present detailed ablations on the number of random features for pooler experiments on the Glue tasks (see figure 8).

## J   ADDITIONAL EXPERIMENTS

We show some additional experiments with different SNNK layers on various tasks.

## J.1   POOLER EXPERIMENTS

In this section, we present additional pooler experiments using a linearization of the tanh kernel as well as pooler experiments for vision transformers.

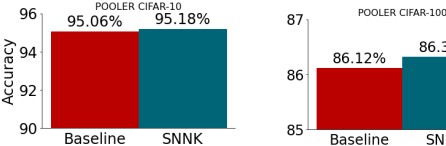

Figure 10: Results on linearized pooler experiments on CiFar-10 and CiFar-100.

### J.1.1 Experiments using Brute Force Linearization of Tanh

Since tanh is the activation function used in pooler layers, we compute the linearization of the tanh layer via the method sketched in Appendix L.1. The sparse features produced by this kernel makes learning difficult and produces poor performance (see table 3). This result motivates the need for novel techniques that is developed in this work.

Table 3: Tanh-SNNK experiments on some GLUE benchmarks. The baseline numbers for BERT-base are from (Lee et al., 2019). MCC score is reported for CoLA, F1 score is reported for MRPC, Spearman correlation is reported for STSB, and accuracy score is reported for RTE. All results are an average of 5 seeds.

| Dataset | RTE | MRPC | STSB | COLA |
|---|---|---|---|---|
| Bert-baseline | 57.5 | 81.5 | 78.1 | 29.4 |
| Tanh-SNNK-pooler | 52.08 | 68.71 | 63.49 | 20.72 |

### J.1.2 Linearizing the Pooler Layer in Vision Transformers

Vision Transformers introduced by (Kolesnikov et al., 2021) introduces a [CLS] token which can be used for image classification. Like BERT, ViT also has a pooling layer (a linear layer with tanh activation) that is applied to the representation of the [CLS] token coming from the final transformer layers. One can similarly use that representation for image classification. In that setting, we apply our linearization of the pooler layer. Our method matches the baseline accuracy on Cifar 10 and outperforms on Cifar-100 (see figure 10).

### J.2 SNNK-Adapter Experiments

We present additional experiments on GLUE benchmarks using a Cosine-SNNK adapter layer. Even though sinusoid activation is not common in Transformers or adapters, we see that they perform relatively well beating the baseline on 4 out of 8 tasks while using only $1/3$ of the training parameters.

Table 4: Cosine-SNNK experiments on GLUE benchmarks. The adapter baseline numbers are from (Moosavi et al., 2022). MCC score is reported for CoLA, F1 score is reported for MRPC and QQP, Spearman correlation is reported for STSB, and accuracy scores are reported for the other tasks. All results are an average of 5 seeds.

| Dataset | RTE | MRPC | QNLI | QQP | SST-2 | MNLI | STSB | COLA |
|---|---|---|---|---|---|---|---|---|
| Bert-Adapter-baseline | 63.83 | 84.8 | **90.63** | **88.12** | 91.74 | **83.53** | 88.48 | **56.51** |
| Cosine-SNNK-Adapter | **64.77** | **85.06** | 88.67 | 86.471 | **91.97** | 79.28 | **88.57** | 52.12 |

### J.3 Training a Bundled Network

We present proof-of-concept experiments on training using a *bundled* network. We use the setting of finetuning a BERT-based model where the pretrained transformer layers are frozen and only the pooler and classifier layers are tuned.

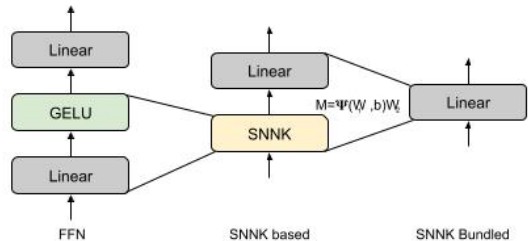

Figure 11: FFN and SNNK adapted FFN layer of BERT and ViT.

We linearize the pooler layer and combine it with the classifier layer to train a matrix $\overline{W}$ of size (# random features $\times$ # classes), resulting in a dramatic drop in training parameters. We maintain competitive performance while using $1024$ random features thus training approximately $1/30$ of the number of parameters of the baseline model which trains both the pooler and classifier head. We also train a linear probe, where the pooler is also frozen and only the classifier head is trained. The results are presented in table 5.

We would like to emphasize that these results are remarkable, particularly on large datasets like MNLI, where we are only about 2.7 points off the top row despite being about $1/30$th smaller. Our results also show that the BERT representations generally can not linearly separate the downstream classes. Our novel methods introduce non-linearities without many additional parameters allowing for implicit construction of complicated decision boundaries. Moreover, our bundled networks can be trained *stably* at very high learning rates up to 2e-2, allowing them to converge rapidly.

STSB dataset is a regression task and as explained in sec 3.3, we have a closed form formula for $\overline{W}$ in that case. We computed the features of the BERT vectors via our random feature mechanism and calculated the coefficients of the linear regressor using the closed form formula, which gives us a Spearman correlation of 67. The entire fitting and evaluation takes barely a few seconds on a CPU. We get the same result when we import the linear regression function from *sklearn*.

On Cifar-10 and CiFar-100, we achieve an accuracy of 95.8 and 82.73 respectively, closely matching the baseline results.

Table 5: Bundled-SNNK experiments on GLUE benchmarks. MCC score is reported for CoLA, F1 score is reported for MRPC and QQP, Spearman correlation is reported for STSB, and accuracy scores are reported for the other tasks. All results are an average of 5 seeds.

| Dataset | # Training Parameters | RTE | MRPC | QNLI | QQP | SST-2 | MNLI | STSB | COLA |
|---|---|---|---|---|---|---|---|---|---|
| Pooler + Classifier (Lee et al., 2019) | $\sim 59k$ | 57.5 | **81.5** | **74.5** | **72.0** | **84.9** | **56.4** | **78.1** | 29.4 |
| Linear Probe | $\sim 1k$ | 57.76 | 81.23 | 69.18 | 62.22 | 83.02 | 43.4 | 69.87 | **35.41** |
| Bundled-final-layers (ours) | $\sim 2k$ | **60.01** | 81.48 | 72.21 | 70.19 | 82.97 | 53.69 | 71.84 | 32.79 |

### J.4 UPTRAINING TRANSFORMERS

In this section we provide experimental details and results on how SNNKs can be used to replace certain FFN blocks in Transformers. It is well-known that the FFN blocks account for more than half of the training parameters and the storage size. If $d$ is the hidden size of the transformer, the FFN block consists of 2 linear layers, one taking the input representation of dimension $d$ to $4d$ with GELU non-linearity and the other taking the intermediate representation down to dim $d$. Our aim here is to replace the first expansion linear layer with non-linearity by a small SNNK layer employing only **8** random features (Figure 11). After we have trained a model, we can bundle the SNNK with the following linear layer entirely bypassing these large FFN blocks. In particular, we replace $f(x) = Gelu(xW_1 + b_1)W_2 + b_2$, where $W_1, W_2^T \in \mathbb{R}^{d \times 4d}$ with $f'(x) = \phi(x)A^T W_2 + b_2$, where $A = \psi(W_1, b_1)$. In our settings $d = 768$ and number of random features is 8. Thus $A$ is of size $8 \times 4 * d$. As described by our bundling process, for forward pass, $A^T W_2$ can be precomputed into a matrix of size $d \times 8$ to save further memory and computational cost.

For both image and sequence classification tasks, we replace the top $k$ FFN layers of BERT and ViT models by SNNKs using the procedure described above. Each successive replacement of FFN layer with SNNK leads to reduction of both number of parameters and flop count (see Table 6). The flops

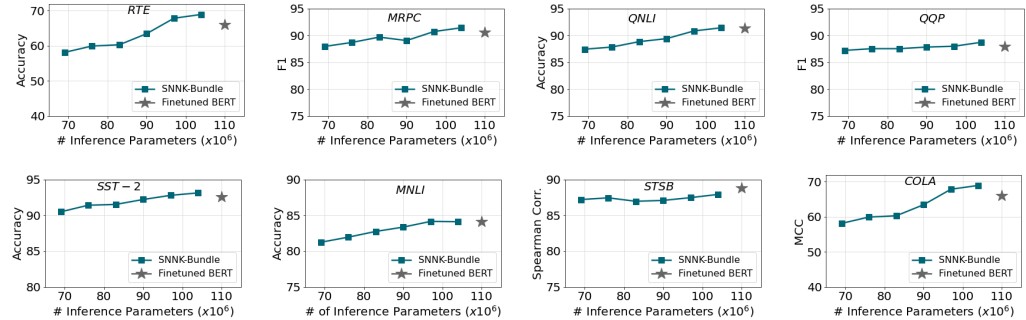

Figure 12: Uptraining experiments using BERT on GLUE benchmark. MCC score is reported for CoLA, F1 score is reported for MRPC and QQP, Spearman correlation is reported for STSB, and accuracy scores are reported for the other tasks. Full column corresponds to full finetuning and the BERT results are sourced from (Pfeiffer et al., 2020)

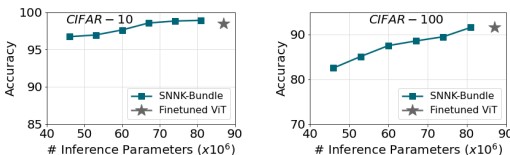

Figure 13: ViT uptraining results on CIFAR-10 and CIFAR-100.

are counted using an batch size of 64 with a sequence length of 128 for BERT models and using a batch of 32 images for the ViT models. Note that we can substantially reduce the size/flop counts during inference due to our novel bundling process. In fact, with 6 layers bundled, the BERT size is **226**.**71** MB down from 440, and for ViT, it is **176**.**42** Mb down from 346 Mb (almost a *50%* reduction). We plot the change in accuracy as the model size and flops decrease Figure 12. The notable reduction in model size and FLOPs comes with only a slight decrease in accuracy.

On a free Google Colab, running a batch size of 64 with a sequence length of 128, it takes our SNNK-BERT .32 secs compared to .44 seconds to the original BERT model. For SNNK-ViT, it takes .26 seconds to run inference on a batch of 32 images whereas ViT takes .33 seconds.

Table 6: Detailed analysis of parameters and flops of bundled transformers.

| Model | | Full | 12 | 11 | 10 | 9 | 8 | 7 |
|---|---|---|---|---|---|---|---|---|
| BERT | # Training parameters (millions) | 110 | 107 | 100 | 93 | 86 | 79 | 72 |
| | # Inference parameters (millions) | 110 | 104 | 97 | 90 | 83 | 76 | 69 |
| | # Training Flops (billions) | 716 | 697 | 678 | 659 | 640 | 620 | 601 |
| | # Inference Flops (billions) | 716 | 677 | 639 | 600 | 561 | 523 | 484 |
| ViT | # Training parameters (millions) | 86 | 84 | 77 | 70 | 63 | 55 | 48 |
| | # Inference parameters (millions) | 86 | 81 | 74 | 67 | 60 | 53 | 46 |
| | # Training Flops (billions) | 563 | 548 | 533 | 519 | 504 | 489 | 475 |
| | # Inference Flops (billions) | 563 | 533 | 503 | 474 | 444 | 414 | 384 |

## K  DETAILED COMPARISONS WITH ADDITIONAL BASELINES

We present detailed comparisons our SNNK-inspired adapter methods with different SOTA parameter efficient finetuning (PEFT) methods. In table 7, results for BiTFit, Adapter, AdapterDrop, LoRA, Transformer-probing, LoRA-Fix, LayerNorm Tuning, LePE Tuning, RPB Tuning, KAdaptation are from He et al. (2022b), Visual Prompt Tuning results are from Jia et al. (2022) and that of AdaptFormer are from Chen et al. (2022). Adapter (Houlsby) is a baseline trained by us that uses the Houlsby configuration as described in (Pfeiffer et al., 2020). Our SNNK-inspired adapters perform competitively amongst various parameter efficient methods. RELU-SNNK-Adapter* model has a pooler layer but it is kept frozen, while RELU-SNNK-Adapter does not use a pooler layer as is commonly the case with ViTs.

Table 7: Comparisons with SoTA parameter efficient methods for Vision Models.

| Methods | # Training Parameters (in millions) | Cifar- 10 | Cifar-100 |
|---|---|---|---|
| Full Finetune | 85 | 98.95 | 91.67 |
| BitFit | .36 | 92.3 | 81.0 |
| Adapter (Houlsby) | 1.8 | 99.1 | 91.2 |
| Adapter | 1.51 | 98.4 | 90.6 |
| AdapterDrop | .17 | 96.8 | 88.4 |
| LoRA | .22 | 98.7 | 90.6 |
| Transformer-probing | 3.2 | 96.5 | 86.9 |
| LoRA-Fix | .15 | 96.2 | 88.3 |
| LayerNorm Tuning | .08 | 92.2 | 71.7 |
| Attention Tuning | 28.41 | 93.9 | 85.7 |
| LePE Tuning | .17 | 93.7 | 90.8 |
| RPB Tuning | .15 | 96.7 | 87.0 |
| KAdaptation | .11 | 97.9 | 91.2 |
| Visual Prompt Tuning | .08 | - | 90.97 |
| AdaptFormer-64 | 1.26 | - | 91.86 |
| ReLU-SNNK-Adapter* | .3 | 98.87 | 92.11 |
| ReLU-SNNK-Adapter | .3 | 98.2 | 91.1 |

We now present baselines for PEFT methods for the GLUE tasks. The training parameters in the table refer to the additional training parameters injected to the frozen transformer models. Bit-Fit results are taken from Zaken et al. (2022), Adapter (Houlsby and Pfeiffer) from Pfeiffer et al. (2020). For the Lora baseline, we ran our own experiments using the same Lora hyperparameters for Roberta-base (Hu et al., 2022).

Table 8: Comparisons with SoTA parameter efficient methods on GLUE benchmarks. MCC score is reported for CoLA, F1 score is reported for MRPC and QQP, Spearman correlation is reported for STSB, and accuracy scores are reported for the other tasks. Final classifier parameters are not used for counting the total number of training parameters.

| Dataset | # Training Parameters (in millions) | RTE | MRPC | QNLI | QQP | SST-2 | MNLI | STSB | COLA |
|---|---|---|---|---|---|---|---|---|---|
| Full Finetune | 110 | 66.2 | 90.5 | 91.3 | 88.3 | 92.6 | 84.1 | 88.8 | 59.5 |
| Adapter-baseline (Moosavi et al., 2022) | .9 | 63.83 | 84.8 | 90.63 | 88.12 | 91.74 | 83.53 | 88.48 | 56.51 |
| Adapter (Houlsby) | 1.8 | 69.8 | 91.5 | 91.2 | 88.6 | 92.8 | 84.1 | 89.2 | 59.1 |
| Adapter (Pfeiffer) | .9 | 70.8 | 89.7 | 91.3 | 88.2 | 90.2 | 84.1 | 89.0 | 58.9 |
| Adaptable Adapter (Moosavi et al., 2022) | - | 64.25 | 85.09 | 89.96 | 88.09 | 91.31 | 82.89 | 88.25 | 51.44 |
| LoRA | .3 | 72.5 | 90.1 | 91.5 | 88.5 | 92.7 | 84.1 | 89.1 | 59.2 |
| BitFit | .1 | 72.3 | 90.4 | 90.2 | 84.0 | 92.1 | 82.2 | 89.2 | 58.8 |
| Relu-SNNK-Adapter | .3 | 69.68 | 91.26 | 90.44 | 85.82 | 92.31 | 82.06 | 88.81 | 58.21 |

## L USING POLYNOMIAL KERNELS FOR BRUTE FORCE CONSTRUCTION OF RANDOM FEATURES

Kar & Karnick (2012) constructs random features for kernels of the form $K(\boldsymbol{x}, \boldsymbol{y}) = f(\boldsymbol{x}\boldsymbol{y}^\top)$, where $f$ has a power series expansion of the form $f := \sum a_n x^n$, and $a_n \geq 0$. This can be easily extended to the case where $f$ does not necessarily have positive Taylor coefficients. Write $f$ as $f := f_1 - f_2$, where $f_1$ and $f_2$ have positive coefficients, and $K(\boldsymbol{x}, \boldsymbol{y}) = K_1(\boldsymbol{x}, \boldsymbol{y}) - K_2(\boldsymbol{x}, \boldsymbol{y})$. Each $K_i$ admits a random feature mechanism $\Phi_i$ and thus $K(\boldsymbol{x}, \boldsymbol{y}) \sim [\Phi_1(\boldsymbol{x})|\Phi_2(\boldsymbol{x})][\Phi_1(\boldsymbol{y})| - \Phi_2(\boldsymbol{y})]^\top$ where $|$ refers to concatenation of 2 vectors along their columns. The approximation error can be bounded by the sum of the approximation errors for each factor. We refer to this construction as the brute force variant.

### L.1 EXTENDING POLYNOMIAL KERNELS TO TANH

Tanh is an odd function so $a_n = 0$, if n is even. Moreover, the non-zero terms alternate in sign, i.e.

$$a_n \begin{cases} > 0, & \text{if } n \equiv 1 (\text{mod } 4) \\ < 0, & \text{if } n \equiv 3 (\text{mod } 4) \\ = 0 & \text{otherwise} \end{cases}$$

Splitting it up in 2 power series with the positive and negative terms produces extremely sparse kernels $\Phi_i$, and this is due to the external measure used in (Kar & Karnick, 2012). Even with rejection sampling, i.e. essentially sampling only odd terms still produces sparse kernels. This becomes a bottleneck in the pooler experiments where most of the accuracies are contributed by the pretrained features and applying a sparse kernel distorts them. We use the same random Rademacher matrices for construction of both $K_1$ and $K_2$ as it reduces the variance (Choromanski et al., 2022).

Table 3 shows some results using the above kernel for the pooler experiments. The results are considerably worse for larger datasets. This negative results show the need for developing novel techniques discussed in this paper.

## M FOURIER TRANSFORMS OF SOME COMMONLY USED ACTIVATION FUNCTIONS

In this section we compute the Fourier Transform of some well-known activation functions.

### M.1 SINE

Sine activation was used by Sitzmann et al. (2020). This is well-known and is given by :

$$FT[\sin(x)](k) = \frac{i}{2}[\delta(k + \frac{1}{2\pi}) - \delta(k - \frac{1}{2\pi})] \tag{24}$$

where $\delta$ is the Dirac delta distribution.

### M.2 COSINE

The Fourier transform of cosine is well known and is given by

$$FT[\cos(x)](k) = \frac{1}{2}[\delta(k + \frac{1}{2\pi}) + \delta(k - \frac{1}{2\pi})] \tag{25}$$

Note that the Fourier transform is real as cosine is an even function.

### M.3 RELU

The Fourier transform of RELU function is not well behaved. So one can use a smoothed version of RELU and compute it's Fourier transform instead. This method is employed in (Bresler & Nagaraj, 2020). In fact our method can be seen as a generalization of the results presented in the above paper.

## N   LIMITATIONS

The key ingredient in the computation of URFs is the computation of the Fourier Transform (FT) of the activation function. However, the FT of some of the activation functions used in practice is not well-behaved, e.g. ReLU, see (Bresler & Nagaraj, 2020) for the derivation of the FT of a "smooth" truncated ReLU. Smoothing and truncating the function incurs an error.

Furthermore, approximating regular feedforward layers with random feature techniques incurs other errors that might propagate from one SNNK layer to the next SNNK layer (see Section B for a detailed discussion).

As we can see from Section J.4, replacing multiple layers of FFL with SNNK results in a trade-off between accuracy and efficiency. This trade-off is a limitation of our current work and an open question for future research.

Even though the approximation of the kernel may depend on the $L^1$ integrability/smoothness of the activation function, we noticed that in practice taking a proxy function to simulate the kernel and learning the weights work well as long as they are not ill-conditioned (i.e. not too spiky or sparse).

