# OpenReview forum: "Scalable Neural Network Kernels"
_ICLR.cc/2024/Conference — ICLR 2024 poster_

### Official Review · Reviewer_hP8B · 2023-10-27

**Soundness:** 3 good
**Presentation:** 3 good
**Contribution:** 3 good
**Rating:** 5
**Confidence:** 4

**Summary:**

This paper proposes to approximate feedforward layers with kernels, thereby achieving better computational efficiency and sometimes better accuracy. This paper empirically verifies their claim on many vision and language datasets over various architectures.

**Strengths:**

This paper is well-written, easy to follow, and empirical results seem strong.

**Weaknesses:**

The main weakness of the paper is that I am not convinced by whether SNNK can practically replace feed forward layers in practice. See Questions for more details

**Questions:**

1. This paper talks a lot about achieving computational efficiency through dimensionality reduction (if m << d). Could I achieve the same effect by using a feed forward layer but simply reducing the latent dimension from d to m?
2. Could the authors share empirical evidence that replacing feedforward layers with SNNK indeed results in faster training?
3. In my understanding if SNNK were to replace feed forward layers then there should be an experiment whether the authors replace every feed forward network in Transformers with SNNK and report the results?
4. Is there any particular reason as to why the SNNK adapted architectures in Figure 2 look the way they are? In other words, how might one understand the interplay between SNNK and feed forward layers in the same architecture?

---

> ### Author Response · Authors · 2023-11-17
> **Response to Reviewer**
>
> We thank the reviewer for their insightful reviews.
>
> **Can SNNK practically replace feed forward layers in practice ?**
>
> We thank the Reviewer for this excellent question. We have shown settings where we can reliably replace feedforward layers. In fact, in Figure 12, 13, we show the performance, as we replace feedforward layers in Transformers via SNNKs. This matter, at the end of the day, boils down to a trade-off between efficiency/compression vs accuracy (table 6).
>
>
> **SNNKs vs methods that decrease the dim of the hidden layers:**
>
> We thank the reviewer for this question. We have the experiment for UCI where we compared SNNK with a three layer MLP and we changed the middle layer size to vary the number of parameters of MLP. We find that at the same parameter size (figure 6),  SNNKs perform better than MLPs with reduced dimensions.
>
> Commonly known methods that can reduce the dimension of the hidden layers (by minimizing the quality loss, as opposed to brute-force layer-size reduction techniques that affect accuracy) are various pruning techniques. In pruning methods, one removes neurons which have small values creating a dense sub-network. However, our method is completely different from pruning as we never remove neurons, rather create a dense representation of the weights via Random Features. Another main difference from pruning methods, which mostly focus on the weights of the network, is that we create a dense projection of the inputs as well.
>
> We added a line in the introduction contrasting our method with various techniques like distillation, pruning and quantization.
>
>
> **Empirical evidence that replacing feedforward layers with SNNKs results in faster training:**
>
>
> We conducted experiments demonstrating that during inference, SNNK-based transformer models exhibit lower memory usage and faster inference compared to the original model. Additionally, during training, we observed a reduced memory footprint. These findings are detailed in Section J.4 of the appendix, where we included results showcasing the replacement of FFN with SNNKs.
>
> To summarize our findings : We show that with 6 layers bundled, the BERT size is 226.71 MB down from 440, and for ViT, it is 176.42 Mb down from 346 Mb (almost a 50% reduction). We also see that on a free Google Colab, running a batch size of 64 with a sequence length of 128, it takes our SNNK-BERT .32 secs compared to .44 seconds to the original BERT model. For SNNK-ViT, it takes .26 seconds to run inference on a batch of 32 images whereas ViT takes .33 seconds. Table 6 provides details of how much increase each additional layer replacement gives in efficiency.
>
> **Replacing every feedforward network in Transformers with SNNKs:**
>
> Thank you very much for the comment. We want to emphasize that in order to achieve substantial computational gains, **one does not necessarily need to remove all the feedforward layers**. It is important to emphasize this since, each SNNK approximating a regular feedforward layer introduces an error and that error accumulates over the deep network from layer to layer (see our theoretical discussion above and also Appendix sec B).
>
> Having said that, as requested, we have conducted additional detailed ablation studies over SNNKs replacing varying numbers of the feedforward layers and present the results below (Please see section J.4 for more details).  If $d$ is the hidden size of the transformer, the FFN block consists of 2 linear layers, one taking the input representation of dimension $d$ to $4d$ with GELU non-linearity and the other taking the intermediate representation down to dim $d$. Our aim here is to replace the first expansion linear layer with non-linearity by small SNNK layers employing only 8 random features. We see from Table 6 about ~40-50% gain in efficiency while from Figure 12 and 13 we see that the tradeoff with accuracy is minimal.
>
>
> **Is there any particular reason as to why the SNNK adapted architectures in Figure 2 look the way they are? In other words, how might one understand the interplay between SNNK and feed forward layers in the same architecture?**
>
> Thank you for the excellent question. Figure 2a aligns with our formulation, while the rationale behind the design choice for Figure 2b is detailed in Appendix section D. Regarding MLP (Figures 2c and 2d), we tested several configurations and presented results for the most effective ones. In the case of adapter experiments (Figures 2d and 2e), we adhered to the design outlined in [1].
>
> [1] Parameter-Efficient Transfer Learning for NLP Neil Houlsby, et al. ICML 2019

---

> ### Author Response · Authors · 2023-11-20
> **updated version of the paper and addressed comments**
>
> Dear Reviewer hP8B,
>
> We would like to once more thank you for your review. We have addressed all the questions and provided corresponding edits in the manuscript (see: updated version). We have conducted an extensive set of additional experiments, in particular accurately measuring the quality of hybrid SNNK-FFL architectures (in terms of the model's accuracy) for configurations with different number of SNNKs and feedforward layers. Furthermore, we have conducted experiments with larger datasets, confirming all our previous findings (see our general comment: "Summary of the Changes", for the overview of the changes made in the manuscript). Thus we would sincerely appreciate if the Reviewer considers raising the score.
>
> Yours sincerely,
>
> The Authors

---

> > ### Comment · Reviewer_hP8B · 2023-11-21
> > **Response to Rebuttal**
> >
> > I sincerely thank the authors for their hard work in producing the rebuttals. I am afraid I am not able to offer any constructive comments at this point. I will keep my score, and leave it for the area chair to decide whether this paper is an interesting contribution.

---

### Official Review · Reviewer_YEib · 2023-10-27

**Soundness:** 2 fair
**Presentation:** 3 good
**Contribution:** 2 fair
**Rating:** 5
**Confidence:** 3

**Summary:**

This paper introduces scalable neural network kernels (SNNKs), which disentangle the inputs and parameters of a feedforward layer before connecting them via a dot product kernel. The key ideas are:

- SNNKs approximate regular feedforward layers but with reduced parameters by replacing the weight matrix with a low-dimensional embedding. This allows compression of the layer.

- They introduce universal random feature maps to instantiate different SNNK variants based on the Fourier transform of the activation function.

- SNNKs can express more complex relationships beyond standard feedforward layers. They demonstrate this with a ReLU-SNNK layer related to arc-cosine kernels.

- SNNKs enable a neural network bundling process to compactify model architectures. In the extreme case, the entire network can be expressed as a two-tower computation.

- For certain losses like MSE, the optimal parameters of a fully bundled network can be solved in closed form, bypassing backpropagation.

- Experiments validate SNNKs on tasks ranging from kernel approximation to Transformer fine-tuning. SNNK adapters match baseline accuracy with 5x fewer parameters.

- Bundled SNNK models maintain accuracy while reducing parameters 30x. Closed-form solutions for regression produce strong results.

In summary, the paper provides a thorough theoretical analysis of SNNKs along with empirical validation. The ideas open interesting research directions in model compression, faster training, and expressive power beyond standard neural network layers.

**Strengths:**

Here are some of the main strengths of this paper:

- It makes an insightful connection between scalable kernel methods and neural network layers, introducing a novel perspective on feedforward layers.

- The concept of SNNKs is very clearly presented along with detailed theoretical analysis and constructions.

- The Fourier transform based universal random feature mechanism to instantiate SNNKs is interesting and useful.

- SNNKs provably increase expressive power over standard layers, as shown through the analysis of the ReLU-SNNK layer.

- The neural network bundling process enabled by SNNKs is an impactful idea for model compression and acceleration.

- The paper provides extensive empirical validation ranging from synthetic data to large Transformer models across vision and language.

- Both model compression and training acceleration are demonstrated convincingly through the experiments.

- The writing is clear, incremental, and easy to follow. Theoretical concepts are explained intuitively.

Overall, the solid theoretical foundation, novel perspectives introduced, and thorough experimentation are major strengths. The paper makes well-motivated connections between areas leading to useful techniques for efficient deep learning.

**Weaknesses:**

Some potential weaknesses or limitations of this paper:

- The focus is on feedforward fully-connected layers, not convolutional or recurrent layers commonly used in modern networks.

- Experiments are limited to standard datasets and models; more complex domains like bioinformatics are not evaluated.

- There is no investigation into how SNNKs affect representation learning or generalization. The emphasis is on compression.

- Optimization and learning dynamics with SNNKs are not analyzed, apart from the fully bundled case.

- The work does not connect to broader topics like kernel methods or metric learning.

- Ablation studies teasing apart the contributions of different components could be more detailed.

- The writing in parts of the theory and experiments lacks clarity or intuitive explanations.

- Practical guidance on hyperparameter selection and tuning SNNKs is limited.

- Applications beyond efficiency, like using SNNKs for privacy or interpretability, are not explored.

- Potential negative results or limitations of SNNKs compared to standard layers are not discussed.

In summary, the lack of experiments on more complex data and models, limited analysis of learning dynamics, and minimal connections to related areas are notable weaknesses. However, the paper makes excellent contributions within its defined scope.

**Questions:**

Please comment on the issues raised in the weaknesses part.

---

> ### Author Response · Authors · 2023-11-17
> **Response to Reviewer : Part-I**
>
> We are happy the Reviewer admits that the paper provides a thorough theoretical analysis along with extensive empirical verification and that the ideas open interesting research direction in model compression, faster training and expressive power beyond standard NNs.
>
> **The focus is on feedforward fully-connected layers, not convolutional or recurrent layers commonly used in modern networks.**
>
> We would like to emphasize that feedforward layers are some of the building blocks of virtually all existing neural networks, including RNNs, CNNs and Transformers. In RNNs, feedforward layers are used in particular to define the gating mechanism. In CNNs, they often constitute the last layers of the models (containing most of the parameters of the model). Thus SNNKs have an impact also on those architectures. In this paper, we have focused particularly on Transformers, since they became new SOTA also in domains, where RNNs or CNNs were traditionally applied (e.g. NLP and vision).
>
> **Experiments are limited to standard datasets and models; more complex domains like bioinformatics are not evaluated.**
>
> Thank you for the question. Following Reviewer’s comment, we have conducted additional experiments, **confirming all our previous findings**. In particular, we have added experimental results on a benchmark dataset like ImageNet (figure 4 right) as well as additional uptraining experiments (where we remove a certain number of MLP layers) in transformers (section 4.4 and Appendix section J.4).
>
>
> **There is no investigation into how SNNKs affect representation learning or generalization. The emphasis is on compression.**
>
> Thank you very much for this interesting question. We would like to emphasize that the considered setting, where SNNKs are applied in the fine-tuning stage, providing computational benefits, is critical for the majority of the applications of Transformers leveraging already pre-trained models. Thus we do think that this scenario provides the most direct positive impact to the Machine Learning community. Needless to say, testing SNNKs for pre-training foundational models and testing  generalization properties of those models is definitely a relevant direction for future work.
> We do believe that SNNKs might indeed have a positive impact on the generalization properties of neural networks since they lead to more compact neural network architectures (while preserving high quality). The compactification of the number of parameters might alleviate the overfitting problem and indeed, this direction was recently explored in the context of adapters (providing by definition compact parameterization) and Robotic controllers (see: for instance RoboAdapter project https://sites.google.com/view/robo-adaptersv).
>
> It is also worth emphasizing that the randomness coming from the random feature mechanism of SNNKs might also have a positive impact on generalization, as leading to training more robust models (if for example different phases or iterations of training apply freshly sampled sets of projections). The phenomenon of robustifying the models via various randomization strategies in training is a large area of research and SNNKs naturally provide such a mechanism. As mentioned above, we leave detailed analysis of the generalization properties of SNNKs to future work.
>
> **Optimization and learning dynamics with SNNKs are not analyzed, apart from the fully bundled case.**
>
> Thank you very much for the question. We would like to sincerely ask the Reviewer to clarify what they mean by the “optimization and learning dynamics with SNNKs”. Our extensive empirical verification clearly shows that SNNK-based models can be efficiently trained with standard optimization methods across a wide range of tasks (involving both: Finetuning Transformers, Uptraining Transformers and attention-free neural networks) and thus do not require any customized optimization techniques.

---

> > ### Author Response · Authors · 2023-11-17
> > **Response to Reviewer : Part-II**
> >
> > **The work does not connect to broader topics like kernel methods or metric learning.**
> > We sincerely thank the Reviewer for the question. In fact, SNNKs **have a direct connection with kernel methods** and even more, were directly inspired by the scalable kernel methods leveraging random feature map techniques such as: “Random Features for Large-Scale Kernel Machines” by Rahimi & Recht. We explicitly state it in the paper several times, in particular as early as in the second paragraph of the introduction and then in the first two paragraphs and the last paragraph of the Related Work section (where we discuss: (1) Johnson Lindenstrauss Transform which is a random feature based technique for the dot-product kernel and (2) random feature map techniques for the approximation of the softmax kernel and arc-cosine kernels). The proposed mechanism of URFs can be thought of as a new class of random features and can be applied in regular kernel methods to scale them up. It also utilizes as one of its key components the softmax-kernel linearization technique from: “Chefs’ Random Tables: Non-Trigonometric Random Features” by Likhoshertov et all. (2022), as we clearly state in the paper above Eq. 8.
> >
> > One of the intriguing properties of SNNKs is that they are also strictly more expressive than regular feedforward layers, as allowing to model complicated relationships beyond the functions of the dot-products of parameter-input vectors. We emphasize it as early as in the abstract and then in the Introduction, where we explicitly list it as one of our main contributions and put in bold font (third point on the contributions’ list). In the corresponding point of the bullet list given there, we already provide an insight by connecting this observation with the class of the so-called arc-cosine kernels. We then provide detailed connections in the dedicated Sec. 3.2., where we present in particular the theorem on scaling up kernel methods leveraging arc-cosine kernels.
> > Finally, in the first set of experiments in the paper we focus on pointwise kernel estimation (Sec. 4.1) with random features which are experiments used on the regular basis to evaluate the quality of the scalable kernel methods leveraging random features (see for instance: “Orthogonal Random Features” by Xu et al., first paragraph of Sec. 5).
> >
> > **Ablation studies teasing apart the contributions of different components could be more detailed.**
> >
> > Thank you so much for this question. The main hyperparameter in SNNK is the number of random features and we have shown that performance is directly correlated with the number of random features (Figure 5, 8). Regarding other architectural choices, we have only used SNNKs as drop in replacements for various modules.
> > Moreover, we show the importance of starting from pretrained weights, instead of random weights in the pooler fine-tuning setting (see Figure 9 (right)).
> >
> > **The writing in parts of the theory and experiments lacks clarity or intuitive explanations.**
> >
> > Thank you very much for the comment. We have added an additional section in the Appendix (Section A), providing additional intuition regarding the theory. We would like to sincerely ask the Reviewer which parts of the empirical section are not clear enough and we are very happy to conduct the corresponding edits to address it.
> >
> > **Practical guidance on hyperparameter selection and tuning SNNKs is limited.**
> > We sincerely thank the Reviewer for this question. Details regarding hyperparameter selection are given in the Appendix H. For different experimental settings, we started with the hyperparameters that other authors have used for that dataset and the given base model. After that, we only tuned the learning rate, observing that SNNKs prefer higher learning rates.
> > We can see in figure 5 and 8, the correlation between the number of random features and accuracy. In practice we found that 16 random features work well for most experiments using adapters.
> >
> >  We hope that this clarifies how the hyperparameters are chosen for different experimental settings.
> >
> > **Applications beyond efficiency, like using SNNKs for privacy or interpretability, are not explored.**
> >
> > Thank you for this question. We believe that the potential to reduce certain optimization problems to a convex optimization problem will have applications in differential privacy and machine unlearning (for example [1]). This is relevant for SNNKs since, as we explain in Sec. 3 (last paragraph before Sec. 4), in the fully-bundled setting with squared L^2-norm loss, the optimization problem becomes a regular linear regression.
> >
> > Regarding interpretability, we believe one can use the same interpretability tools like Lime, Shapley values and Integrated Gradients to interpret the SNNKs. We leave such exciting explorations to future work.
> >
> > [1] Certified Data Removal from Machine Learning Models, Chuan Guo, Tom Goldstein, Awni Hannun, Laurens van der Maaten; ICML 2020

---

> > > ### Author Response · Authors · 2023-11-17
> > > **Response to Reviewer: Part-III**
> > >
> > > **Potential negative results or limitations of SNNKs compared to standard layers are not discussed.**
> > >
> > > We thank the reviewer for this question. We have added a corresponding section in the Appendix (Section N) detailing the limitations of our method. To summarize :
> > >
> > > The computation of Universal Random Features (URFs) relies heavily on Fourier Transforms (FTs) of activation functions. But some functions used in practice, like ReLU, do not have well-behaved FTs leading to errors when smoothed or truncated. Approximating feedforward layers with random features can also introduce errors propagating through subsequent layers, impacting accuracy and efficiency. Replacing FFL layers with SNNK involves a trade-off between accuracy and efficiency, posing a limitation and a future research question. Despite relying on the smoothness of activation functions for kernel approximation, using proxy functions and well-conditioned weights generally works in practice, as long as they're not excessively spiky or sparse.

---

> ### Author Response · Authors · 2023-11-20
> **updated version of the paper and addressed comments**
>
> Dear Reviewer YEib,
>
> We would like to once more thank you for your review. We have addressed all the questions and provided corresponding edits in the manuscript (see: updated version). In particular, we have conducted an extensive set of additional experiments and clarified the limitations of the method (see our general comment: "Summary of the Changes", for the overview of the changes made in the manuscript). Thus we would sincerely appreciate if the Reviewer considers raising the score.
>
> Yours sincerely,
>
> The Authors

---

> ### Comment · Reviewer_YEib · 2023-11-21
> **Response to rebuttal**
>
> I'd like to thank the authors for their detailed responses. However, some of concerns still stays the same and therefore I'd like to keep my original score.

---

> ### Author Response · Authors · 2023-11-22
> **Unanswered questions/concerns**
>
> Dear Reviewer,
>
>  Thank you for your response. Would it be possible for you to let us know which of your concerns/questions are not satisfactorily answered so that we can try our best to answer them?
>
>  Yours sincerely,
>
>  The Authors.

---

### Official Review · Reviewer_uu7e · 2023-11-01

**Soundness:** 4 excellent
**Presentation:** 3 good
**Contribution:** 4 excellent
**Rating:** 8
**Confidence:** 3

**Summary:**

The paper introduces Structurally Neural Network Kernels (SNNK), a novel approach to modeling interactions in neural networks. By exploiting the low-rank nature of neural networks, SNNKs offer a significant reduction in parameter count. When used in multilayer perceptrons and transformer models, SNNKs consistently outperformed traditional baselines across multiple datasets, including synthetic data, toy experiments, UCI datasets, GLUE, and CIFAR. A primary advantage is the reduction in storage requirements without compromising accuracy. The work suggests SNNK layers can be integrated seamlessly with other popular techniques, providing both efficiency and enhanced performance, making them a promising tool for neural network architectures.

**Strengths:**

The paper introduces a new computational model, the scalable neural network kernels (SNNK), providing a novel approach to efficient neural network design, particularly for replacing feedforward layers in MLPs.

The design of SNNKs ensures that inputs and parameters are disentangled, leading to efficient final computations via a dot-product kernel, which can greatly reduce computational overhead.

The bundling process highlighted in the paper leads to the compactification of the neural network stack, suggesting potential storage savings and efficiency improvements.

The paper does not rely solely on theoretical claims but provides empirical analysis, spanning from pointwise kernel estimation to practical application scenarios like training Transformers with adapters, strengthening the validity of the proposed methodology.

**Weaknesses:**

The authors should provide some explanation or intuition why their model doesn’t work well in the some of the experiments they have performed.

The analysis of how deep of a feed forward network can be approximated using the proposed method should be analyzed in further details.

Can scalable neural network kernel be applied in any scenario or there are some specific scenarios when SKNN won’t work well. Authors should discuss about such datasets/models. If there is none, then authors should also discuss that. They can include some of the experiments involving ImageNet datasets and other models to further support the claim.

It would be great to analyse how well the SKNN generalizes to the unseen datasets or in general generalizability as compared to normal network network,

**Questions:**

I have mentioned my concerns I the weakness. Authors can go over the points mentioned in the weakness and clarify them.

---

> ### Author Response · Authors · 2023-11-17
> **Response to Reviewer**
>
> We would like to thank the Reviewer for kind words. We are very happy that the Reviewer appreciated the novelty of our approach.
>
> **Providing intuition why “their model doesn’t work in some of the experiments they have performed”:**
>
> We thank the reviewer for this insightful question. On some GLUE datasets, our pooler-SNNK models were underperforming the baseline. Those datasets are some of the largest ones in the benchmark. We believe that reducing the number of training parameters are causing the models to underfit and we observed similar behavior in our adapter experiments as well. We did ablations by increasing the number of random features and we are able to match the baselines (figure 5 & 8) .
>
> **Limitations of the SNNKs:**
>
> As we can see from the sec 4.4 & J.4, replacing multiple layers of FFL with SNNK results in a tradeoff between accuracy and efficiency. This tradeoff is a limitation of our current work and an open question for future research.
>
> We have added a section discussing the limitations of our method (Appendix Sec N). To summarize :
>
> To summarize :
>
> The computation of Universal Random Features (URFs) relies heavily on Fourier Transforms (FTs) of activation functions. But some functions used in practice, like ReLU, do not have well-behaved FTs, leading to errors when smoothed or truncated. Approximating feedforward layers with random features can also introduce errors propagating through subsequent layers, impacting accuracy and efficiency. Replacing FFL layers with SNNK involves a trade-off between accuracy and efficiency, posing a limitation and a future research question. Despite relying on the smoothness of activation functions for kernel approximation, using proxy functions and well-conditioned weights generally works in practice, as long as they're not excessively spiky or sparse.
>
> **ImageNet experiments:**
>
> Thank you very much for the comment. As requested, we have conducted additional experiments, in particular on the ImageNet data. They confirm all our previous findings. In fact on ImageNet, *our SNNK-adapter-ViT achieves a top 1%-accuracy of 83.44, beating the baseline adapter-ViT by .23 (Figure 4 (right)).*
>
> **Generalization of SNNKs to the unseen datasets or as compared to normal NNs:**
>
> Thank you very much for the question. We do believe that SNNKs might indeed have a positive impact on the generalization properties of neural networks since they lead to more compact neural network architectures (while preserving high quality). The compactification of the number of parameters might alleviate the overfitting problem and indeed, this direction was recently explored in the context of adapters (providing by definition compact parameterization) and Robotic controllers (see: for instance RoboAdapter project https://sites.google.com/view/robo-adaptersv).
>
> It is also worth emphasizing that the randomness coming from the random feature mechanism of SNNKs might also have a positive impact on generalization, as leading to training more robust models (if for example different phases or iterations of training apply freshly sampled sets of projections). The phenomenon of robustifying the models via various randomization strategies in training is a large area of research and SNNKs naturally provide such a mechanism. We leave detailed analysis of the generalization properties of SNNKs to future work.

---

> > ### Author Response · Authors · 2023-11-20
> > **Response to Reviewer : Part-II**
> >
> > **The analysis of the impact on the depth of the neural network on the approximation error:**
> >
> > The variance of the estimation of the kernel value $\mathrm{K}(\mathbf{x},\mathbf{y})$ is proportional to $\frac{1}{m}$, where $m$ is the number of random features. This is the case since its estimator can be rewritten as:
> > $$X = \frac{1}{m} \sum_{i=1}^{m} X_{i},$$
> > where each $X_{i}$ provides an unbiased estimation of the kernel value. Since random variables $X_{i}$ are independent, using Azuma’s Inequality, we can also conclude that if $|X_{i}| \leq c$, then:
> >
> > $P[|X-\mathrm{K}(\mathbf{x},\mathbf{y})| \geq \epsilon] \leq 2 \exp\left(-\frac{\epsilon^{2}}{8mc^{2}}\right) .... (1)$
> >
> > for any $\epsilon > 0$.
> >
> > Recall from $\exp(\widehat{\mathbf{x}}^{\top}(\xi)\widehat{\mathbf{w}}(\xi))$ = $\mathbb{E}\_{\mathbf{g} \sim \mathcal{N}(0,\mathbf{I}_{d})} [\Lambda\_{\mathbf{g}}(\widehat{\mathbf{x}})\Lambda\_{\mathbf{g}}(\widehat{\mathbf{w}})]$, where $\Lambda\_{\mathbf{g}}:\mathbb{R}^{d} \rightarrow \mathbb{R}$ is defined as follows for $A \leq 0$:
> >
> > $\Lambda_{\mathbf{g}}(\mathbf{z})$ = $(1-4A)^{\frac{d}{4}}
> >  \exp(A\|\mathbf{g}\|\_{2}^{2}+\sqrt{1-4A}$ $\mathbf{g}^{\top}\mathbf{z}-\frac{\|\mathbf{z}\|\_{2}^{2}}{2}) $
> >
> > Note that the boundedness condition holds if $A<0$ (see Remark 3.1 in the main paper).
> >
> >
> > Now assume that the kernel function $\mathrm{K}$ satisfies (in the region of interest): $|\mathrm{K}(\mathbf{x},\mathbf{w})-\mathrm{K}(\mathbf{u},\mathbf{w})| \leq \delta(a)$, as long as $\|\mathbf{x}-\mathbf{u}\|_{1} \leq a$ for some function $\delta$. Note that this is not a strong assumption as any continuous function on a compact subset of $\mathbb{R}^{n}$ is uniformly continuous, i.e. satisfies the above condition for some $\delta$, where $\delta(a) \rightarrow 0 $ as $a \rightarrow 0$.
> >
> > We then conclude that the probability that the approximate output of the bundled $d$-layer neural network differs from the exact output by at least $\epsilon + \delta(\epsilon) + … + \delta(... \delta(\delta (\epsilon)))$ (composition of $(d-1)$ $\delta$-functions) in the $L^{1}$-norm is upper-bounded by the RHS from the Inequality (1), but with an extra multiplicative factor $d$ (coming from the union-bound). We see that the number of random features needed for an accurate approximation is larger if $\mathrm{K}$ grows faster (e.g. is $L$-Lipschitz with larger constant $L$).
> >
> > We hope that this answers all your questions. Please let us know if you have any more questions for us.

---

### Official Review · Reviewer_zChr · 2023-11-01

**Soundness:** 4 excellent
**Presentation:** 3 good
**Contribution:** 4 excellent
**Rating:** 8
**Confidence:** 3

**Summary:**

The paper introduces Scalable Neural Network Kernels (SNNKs), a novel alternative to regular feedforward layers (FFLs) in neural network architectures. These SNNKs, while approximating the behavior of FFLs, bring in computational advantages by separating the inputs from the neural network's parameters and then connecting them through a dot-product kernel.

The primary contribution is the conceptualization of SNNKs that can mimic FFLs but have better computational attributes. Unlike traditional FFLs, these kernels can capture complex relationships beyond just the functions of the dot-products of parameter-input vectors.

The authors propose a bundling process utilizing SNNKs to condense the architecture of deep neural networks. This leads to compression benefits, and when fully implemented, it results in a bundled network. Interestingly, for specific loss functions like mean squared error, optimal parameters for this bundled network can be explicitly derived, potentially bypassing the need for backpropagation.

An auxiliary outcome of the research is the introduction of a mechanism called "universal random features," which is instrumental in formulating various SNNK variants. This mechanism also holds significance in scalable kernel methods.

The paper goes on to offer a thorough empirical evaluation of the proposed ideas, ranging from point-wise kernel estimation to the fine-tuning of Transformers using new adapter layers inspired by SNNKs. Remarkably, their method achieves up to a 5x reduction in trainable parameters while retaining competitive accuracy.

**Strengths:**

The paper introduces the concept of Scalable Neural Network Kernels (SNNKs), a fresh take on neural network architecture. This novel approach to approximating regular feedforward layers (FFLs) with computational benefits showcases a high degree of originality. The "neural network bundling process" and the notion of a fully bundled network present innovative methods for condensing deep neural network architectures. The "universal random features" mechanism, which aids in the formulation of various SNNK variants, is another original contribution.

The research maintains a high standard of quality, underpinned by a combination of rigorous theoretical foundations and empirical evaluations. Extensive experiments have been conducted across various architectures and datasets, ensuring that the proposed methods are tested in diverse scenarios. The results, especially the reduction in trainable parameters without significant performance losses, stand testament to the quality of the work.

The paper is structured well, with a clear delineation between theoretical concepts, methodologies, and experimental results. While the document is dense with technical details, the authors have made efforts to explain concepts clearly, aided by visual representations where necessary. The inclusion of a comprehensive list of references and contextualization relative to prior work adds to the clarity, helping readers understand the evolution and significance of the presented ideas.

The versatility of SNNKs, as demonstrated by their applicability in various architectures (from PINNs to Transformers), signifies their broad utility. By addressing the computational challenges associated with traditional kernel methods and FFLs, the paper offers solutions that could pave the way for more efficient and scalable neural network models in the future.

**Weaknesses:**

The paper could benefit from a more direct comparison of SNNKs with other existing solutions or methods aimed at network compression or efficiency. Highlighting the unique advantages of SNNKs over these methods would further solidify its significance.

The paper could delve deeper into the robustness of the SNNK approach. Are there scenarios where the approximation might break down? Understanding the edge cases and potential pitfalls would be crucial for practitioners looking to adopt this method.

Providing more explicit details about the implementation, hyperparameters used, or potential challenges faced during the experiments would be beneficial for researchers aiming to replicate or build upon the work.

**Questions:**

The paper mentions that SNNKs can approximate FFLs. Could you provide more insight into the approximation error? In what scenarios might the approximation be suboptimal, and how does the error scale with the depth or complexity of the network?

it would be helpful to understand more about the bundling process's efficiency. How does the network's performance vary as more layers are bundled, especially in deeper architectures?

In the experiments where SNNKs achieved up to a 5x reduction in trainable parameters, were there any notable trade-offs in terms of latency, inference time, or other metrics?

Could you elaborate on the key differences between the "universal random features" mechanism and traditional random feature approaches? What are the primary advantages of this new mechanism?

When applying SNNKs to Transformers, especially in the pooler layer linearization, were there any specific challenges or nuances encountered, given the attention mechanisms and positional encodings in such models?

Given the focus on computational efficiency, were there any hardware-specific optimizations or considerations when implementing SNNKs? How do SNNKs perform across different hardware platforms, like CPUs, GPUs, and TPUs?

---

> ### Author Response · Authors · 2023-11-17
> **Response to the Reviewer: Part-I**
>
> We would like to thank the Reviewer for kind words. We are very happy that the Reviewer appreciated the novelty of our approach.
>
> **Comparing SNNKs with other existing solutions or methods aimed at network compression or efficiency:**
>
> We thank the Reviewer for the interesting question. Some of the most prominent methods that are used for compression or efficiency on a regular basis are various quantization techniques [1,2,3] and distillation methods [4]. Our work is orthogonal to these techniques and can be straightforwardly combined with such methods to provide further gains in compression and model efficiency. We leave this direction to future work.
>
> We have added a couple of lines in the introduction contrasting our work with various compression techniques.
>
> Moreover, we compare our SNNK untrained BERT with DistillBERT [5]. In this setting, we replaced the top k MLPs in BERT with SNNK. Additional experiments along with accuracy/efficiency tradeoffs are detailed in Appendix sec J.4.
>
> | Model                  | Parameters | CoLA  | MNLI  | MRPC  | QNLI  | QQP   | RTE   | SST-2 | STS-B |
> |------------------------|:----------:|-------|-------|-------|-------|-------|-------|-------|-------|
> | DistilBERT             |     66M    | 51.3  | 82.2  | 87.5  | 89.2  | 88.5  | 59.9  | 91.3  | 86.9  |
> | SNNK-BERT top-6 layers |     69M    | 49.86 | 81.23 | 87.92 | 87.41 | 90.56 | 58.12 | 90.48 | 87.19 |
> | SNNK-BERT top-5 layers |     76M    | 56.58 | 81.94 | 88.67 | 87.81 | 90.68 | 59.93 | 91.4  | 87.43 |
>
>
> 1. A Survey of Quantization Methods for Efficient Neural Network Inference; Amir Gholami , Sehoon Kim , Zhen Dong , Zhewei Yao , Michael W. Mahoney, Kurt Keutzer arxiv 2021
> 2. Quantization and Training of Neural Networks for Efficient Integer-Arithmetic-Only Inference; Benoit Jacob, Skirmantas Kligys, Bo Chen, Menglong Zhu, Matthew Tang, Andrew Howard, Hartwig Adam, Dmitry Kalenichenko arxiv 2017
> 3. Understanding INT4 Quantization for Language Models: Latency Speedup, Composability, and Failure Cases; Xiaoxia Wu, Cheng Li, Reza Yazdani Aminabadi,  Zhewei Yao, Yuxiong He ICML 2023
> 4. Knowledge Distillation: A Survey; Jianping Gou, Baosheng Yu, Stephen J. Maybank, Dacheng Tao International Journal of Computer Vision 2021
> 5. DistilBERT, a distilled version of BERT: smaller, faster, cheaper and lighter Victor Sanh, Lysandre Debut Julien Chaumond, Thomas Wolf 2019

---

> ### Author Response · Authors · 2023-11-17
> **Response to Reviewer : Part-II**
>
> **Delving deeper into the robustness of the SNNK approach. Are there scenarios where the approximation might break down? Understanding the edge cases and potential pitfalls would be crucial for practitioners looking to adopt this method.**
>
> We thank the Reviewer for an insightful question. The key ingredient in the computation of URFs is the computation of the Fourier Transform (FT) of the activation function. However, the FT of some of the activation functions used in practice is not well-behaved, e.g. ReLU, see [1] for the derivation of the FT of a “smooth” truncated ReLU. Smoothening and truncating the function incurs an error. Furthermore, approximating regular feedforward layers with random feature techniques incurs other errors that might propagate from one SNNK layer to the next SNNK layer. Even though the approximation of the kernel may depend on the $L^1$ integrability/smoothness of the activation function, we noticed that in practice taking a proxy function to simulate the kernel and learning the weights work well as long as they are not ill-conditioned (i.e. not too spiky or sparse).
>
> We have added the corresponding Limitations section to the Appendix (Sec. N) and another section detailing the error propagation in a bundled deep neural network (Appendix Sec B).
>
>
> **More details regarding implementation, hyperparameters, potential challenges during experiments**
>
> We sincerely thank the Reviewer for this question. Details regarding hyperparameter selection are given in the Appendix H.  For different experimental settings, we started with the hyperparameters that other authors have used for that dataset and the given base model for that setting. The only hyperparameter that we really tuned is the learning rate. The key observation here is the following one : if the training is too slow, increasing the learning rate significantly boosts performance.  In some cases, it may take longer to converge so we trained the models till the validation loss started increasing.
>
> Our code is written in PyTorch. For experiments with pooler and up-training runs, we have built a code on the top of Huggingface Transformers repos. For up-training experiments, we replace the intermediate layer with our SNNK and for the pooler, we replace the pooler by the corresponding SNNK. For adapter experiments, we followed the implementation of Transformer-adapters.
>
> Our entire motivation is to do trade-off between accuracy and efficiency (training and inference). Thus, it was reassuring to see that we can improve performance and batch baselines as we increase the number of random features (figures 5 and 8).
>
> We will make all our implementations publicly available once our manuscript is published.
>
>
> **Clarification regarding experiments with 5x reduction in trainable parameters:**
>
> For these experiments, we showed that we can maintain the same level of accuracy with a certain amount of reduction of trainable parameters. There is a trade-off involved as further reduction of parameters may hurt performance. Figure 12 and 13 shows those trade-offs, i.e. lower inference/storage costs vs accuracy. We have also added more experimental details as well discussion about these tradeoffs in section J.4.
>
>
> **Key differences between URFs and traditional RFs:**
>
> Thank you for your question. Most traditional RF-approaches rely on the Fourier analysis, but via Bochner Theorem and trigonometric functions. In contrast, URFs apply exponential functions. This approach was found to much more efficiently linearize the softmax-kernel, see: Rethinking Attention with Performers, than regular trigonometric random features (and effectively led to trainable linear-attention Transformers, as opposed to the trigonometric variants). Thus one way of thinking about URFs is as an extension of the method presented in that paper, but for a much larger class of kernels.
>
> We have also provided more intuition behind URFs in section A.
>
>
> **Clarification regarding SNNKs for Transformers (pooler layers’ linearization):**
>
> There were no specific challenges involved. We mostly followed the hyperparameters that other authors have applied for pooler tuning. The key insight that we realized is that the SNNKs can be trained with a higher learning rate and need a little bit more time to converge.
>
>
> **Hardware-specific optimization or consideration ? Performance across different platforms:**
>
> No hardware-specific optimization was conducted. We run the experiments on both: GPUs and TPUs. SNNKs are “accelerators-friendly” since their key computational bottleneck is matrix-matrix multiplication.
>
>
> [1] A Corrective View of Neural Networks: Representation, Memorization and Learning Guy Bressler and Dheeraj Nagaraj Conference on Learning Theory 2020.

---

> > ### Author Response · Authors · 2023-11-20
> > **Response to Reviewer : Part-III**
> >
> > **More insight into the approximation error & bundling process’s efficiency:**
> >
> > The variance of the estimation of the kernel value $\mathrm{K}(\mathbf{x},\mathbf{y})$ is proportional to $\frac{1}{m}$, where $m$ is the number of random features. This is the case since its estimator can be rewritten as:
> > $$X = \frac{1}{m} \sum_{i=1}^{m} X_{i},$$
> > where each $X_{i}$ provides an unbiased estimation of the kernel value. Since random variables $X_{i}$ are independent, using Azuma’s Inequality, we can also conclude that if $|X_{i}| \leq c$, then:
> >
> > $P[|X-\mathrm{K}(\mathbf{x},\mathbf{y})| \geq \epsilon] \leq 2 \exp\left(-\frac{\epsilon^{2}}{8mc^{2}}\right) .... (1)$
> >
> > for any $\epsilon > 0$.
> >
> > Recall from $\exp(\widehat{\mathbf{x}}^{\top}(\xi)\widehat{\mathbf{w}}(\xi))$ = $\mathbb{E}\_{\mathbf{g} \sim \mathcal{N}(0,\mathbf{I}_{d})} [\Lambda\_{\mathbf{g}}(\widehat{\mathbf{x}})\Lambda\_{\mathbf{g}}(\widehat{\mathbf{w}})]$, where $\Lambda\_{\mathbf{g}}:\mathbb{R}^{d} \rightarrow \mathbb{R}$ is defined as follows for $A \leq 0$:
> >
> > $\Lambda_{\mathbf{g}}(\mathbf{z})$ = $(1-4A)^{\frac{d}{4}}
> >  \exp(A\|\mathbf{g}\|\_{2}^{2}+\sqrt{1-4A}$ $\mathbf{g}^{\top}\mathbf{z}-\frac{\|\mathbf{z}\|\_{2}^{2}}{2}) $
> >
> > Note that the boundedness condition holds if $A<0$ (see Remark 3.1 in the main paper).
> >
> > Now assume that the kernel function $\mathrm{K}$ satisfies (in the region of interest): $|\mathrm{K}(\mathbf{x},\mathbf{w})-\mathrm{K}(\mathbf{u},\mathbf{w})| \leq \delta(a)$, as long as $\|\mathbf{x}-\mathbf{u}\|_{1} \leq a$ for some function $\delta$. Note that this is not a strong assumption as any continuous function on a compact subset of $\mathbb{R}^{n}$ is uniformly continuous, i.e. satisfies the above condition for some $\delta$, where $\delta(a) \rightarrow 0 $ as $a \rightarrow 0$.
> >
> > We then conclude that the probability that the approximate output of the bundled $d$-layer neural network differs from the exact output by at least $\epsilon + \delta(\epsilon) + … + \delta(... \delta(\delta (\epsilon)))$ (composition of $(d-1)$ $\delta$-functions) in the $L^{1}$-norm is upper-bounded by the RHS from the Inequality (1), but with an extra multiplicative factor $d$ (coming from the union-bound). We see that the number of random features needed for an accurate approximation is larger if $\mathrm{K}$ grows faster (e.g. is $L$-Lipschitz with larger constant $L$).
> >
> > We hope that this answers all your questions. Please let us know if you have any more questions for us.

---

### Author Response · Authors · 2023-11-17
**Summary of the Changes**

We would like to sincerely thank all the Reviewers for their valuable feedback. We address all the comments in more detail below. **We have also conducted a large number of additional experiments** (in particular with larger datasets and SNNKs replacing more feedforward layers), confirming our previous results. We have updated our manuscript (changes are in blue) and summarize our changes below :
- We have conducted additional adapter-SNNK experiments on ImageNet-1k beating the baseline by .23. (see Figure 4 right)
- We have conducted extensive experiments on replacing certain MLP layers in BERT and ViT. We observe a notable reduction in model size and floating-point operations (FLOPs) come with only a slight decrease in accuracy. (sec 4.4 and Appendix sec J.4 )
- We have added sections in the Appendix detailing limitations of our method (Appendix sec N ), details about error propagation over a deep bundled network (Appendix sec B ) and more motivation towards our Universal Random Features (URF) mechanism (Appendix sec A).

---

### Meta-Review · Area_Chair_91LT · 2023-12-05

**Metareview:**

This paper propose scalable neural network kernels (SNNKs) to replace feedforward network in various learning tasks, which improves computational efficiency for the learning procedure. The analysis based on universal random feature brings interesting insights for the proposed scheme. With vibrant reviewers discussion and additional experiments and clarifications that would further improve the paper, I believe it would be interesting for the ICLR audience and suggest accept the paper.

**Justification For Why Not Higher Score:**

Further clarifications and explanations may be needed for the paper to be published

**Justification For Why Not Lower Score:**

The proposed concept is interesting and beneficial to the community to see and think about

---

### Decision · Program_Chairs · 2024-01-16

Accept (poster)